# Defective excitation-contraction coupling and mitochondrial respiration precede mitochondrial Ca²⁺ accumulation in spino-bulbar muscular atrophy skeletal muscle

Caterina Marchioretti[1,2,3,4,16], Giulia Zanetti [1,16], Marco Pirazzini [1,5,16], Gaia Gherardi[1], Leonardo Nogara[1,2], Roberta Andreotti[1,2,3], Paolo Martini [6], Lorenzo Marcucci [1], Marta Canato[1], Samir R. Nath[7], Emanuela Zuccaro[1,2,3], Mathilde Chivet[4], Cristina Mammucari[1,5], Marco Pacifici[1], Anna Raffaello[1,5], Rosario Rizzuto[1], Andrea Mattarei [8], Maria A. Desbats[9], Leonardo Salviati [5,9], Aram Megighian [1,3], Gianni Sorarù[3,10], Elena Pegoraro [10], Elisa Belluzzi [11,12], Assunta Pozzuoli[11,12], Carlo Biz[11], Pietro Ruggieri [11], Chiara Romualdi [13], Andrew P. Lieberman[7], Gopal J. Babu [14], Marco Sandri [1,2], Bert Blaauw[1,2], Manuela Basso [15] & Maria Pennuto [1,2,3,4] ✉

Polyglutamine expansion in the androgen receptor (AR) causes spinobulbar muscular atrophy (SBMA). Skeletal muscle is a primary site of toxicity; however, the current understanding of the early pathological processes that occur and how they unfold during disease progression remains limited. Using transgenic and knock-in mice and patient-derived muscle biopsies, we show that SBMA mice in the presymptomatic stage develop a respiratory defect matching defective expression of genes involved in excitation-contraction coupling (ECC), altered contraction dynamics, and increased fatigue. These processes are followed by stimulus-dependent accumulation of calcium into mitochondria and structural disorganization of the muscle triads. Deregulation of expression of ECC genes is concomitant with sexual maturity and androgen raise in the serum. Consistent with the androgen-dependent nature of these alterations, surgical castration and AR silencing alleviate the early and late pathological processes. These observations show that ECC deregulation and defective mitochondrial respiration are early but reversible events followed by altered muscle force, calcium dyshomeostasis, and dismantling of triad structure.

Spinobulbar muscular atrophy (SBMA, OMIM 313200), also known as Kennedy's disease, is a genetic neuromuscular disease caused by the presence of over 37 expansions of a microsatellite CAG trinucleotide tandem repeat encoding a polyglutamine (polyQ) tract in exon 1 of the androgen receptor (*AR*) gene[1]. SBMA affects 2–5/100,000 subjects, which is probably an underestimation due to misdiagnosis and lack of disease-specific biomarkers. SBMA is a progressive late-onset disease characterized by the selective degeneration of lower motor neurons (MNs) and weakness, fasciculations, and atrophy of skeletal muscle[2]. The disease is androgen dependent, and full manifestations are

exhibited in males. The AR is the main mediator of androgen signaling. Upon androgen binding, the AR undergoes a conformational change that leads to nuclear translocation and activation, which are all necessary steps to induce toxicity, thus explaining the sex-dependent nature of SBMA[3]. The neuromuscular phenotype, sex dependence, and peripheral symptoms are well recapitulated in mouse models of the disease[4–7].

In the last decade, clinical and experimental evidence has established that skeletal muscle atrophy is not only secondary to MN dysfunction but also results from the primary cell-autonomous toxicity of polyQ-expanded AR in myofibers[8–10]. Patients show early muscle pathology and clear signs of myopathy, including elevated levels of serum creatine kinase, myofiber degeneration, and central-core-like areas without mitochondria; all of these changes are detected before or at the onset of clinical symptoms[11–14]. Similarly, signs of myopathy and reduced intrinsic muscle force generation precede spinal cord pathology in transgenic and knock-in SBMA mice[7,15]. Among the primary and secondary pathological processes responsible for muscle atrophy in SBMA, there is evidence that deregulation of androgen signaling is necessary and sufficient to cause functional alterations in the motor unit. Indeed, overexpression of an AR with a normal polyQ tract size selectively in muscle has been found to elicit a phenotype that resembles SBMA in several aspects, showing that AR gain of function in muscle suffices to cause atrophy[16]. In contrast, ablation of polyQ-expanded AR expression selectively in muscle has been found to prevent disease manifestations, unequivocally showing that mutant AR expression in skeletal muscle is necessary to cause an SBMA-like phenotype in mice[17]. Two clinically relevant aspects underline the importance of skeletal muscle in SBMA. First, the levels of blood biomarkers of muscle damage correlate with the severity of the disease[18]. Second, skeletal muscle is a valuable target tissue for therapy development. Indeed, pharmacological intervention to silence polyQ-expanded AR expression with antisense oligonucleotides (ASOs) in peripheral tissues[19], as well as strategies to promote polyQ-expanded AR degradation while stimulating muscle hypertrophy, such as insulin-like growth factor 1 and the beta-agonist clenbuterol[20,21], were all found to ameliorate the phenotypes observed in different SBMA mouse models. These observations support the idea that skeletal muscle is central to SBMA and represents a key target tissue for therapy development. Skeletal muscle accounts for 30–40% of body weight. Pathological processes that occur in skeletal muscle impact the homeostasis of the entire body, thus contributing to the progression/outcome of the disease and offering considerable therapeutic potential, which calls for a better understanding of its pathophysiological basis.

Skeletal muscle contraction is triggered by action potentials in the MN through a process known as excitation–contraction coupling (ECC), which links the electrical events that occur in the sarcolemma with myofiber contraction and force generation through specialized structures known as triads; these structures are invaginations of the sarcolemma that form transversal tubules (t-tubules) that interface with sarcoplasmic reticulum (SR). Fast and sustained muscle contraction requires calcium ($Ca^{2+}$) and adenosine triphosphate (ATP). $Ca^{2+}$ is stored in the lumen of the SR, and ATP is produced by mitochondria located close to the t-tubules. $Ca^{2+}$ links muscle contraction with ATP production. Upon arrival of the action potential at the neuromuscular junction, membrane depolarization leads to a conformational change in dihydropyridine receptors (DHPR) and ryanodine receptors (RYR) that results in $Ca^{2+}$ release from the SR into the myoplasm. The $Ca^{2+}$ concentration in the sarcoplasm is remarkably controlled both in the resting state and during myofiber contraction/relaxation by the combined action of proteins responsible for high buffer capacity, i.e., SR-resident calsequestrin 1 (CASQ1) and CASQ2, sarcoplasmic parvalbumin (PV), and SR $Ca^{2+}$-ATPase (SERCA) 1 and SERCA2 pump $Ca^{2+}$ back into the SR[22]. Another important player in $Ca^{2+}$ dynamics is the SERCA

inhibitor sarcolipin (SLN)[23]. Under physiological conditions, mitochondrial $Ca^{2+}$ regulates oxidative phosphorylation and ATP production[24], and deregulation of $Ca^{2+}$ homeostasis often results in pathological conditions[25]. We have previously shown that polyQ-expanded AR affects muscle metabolism and causes mitochondrial depolarization and mitophagy in the muscle of SBMA mice and patients[5,11,26], and these features were alleviated by ASO-mediated knockdown of polyQ-expanded AR[19].

Using constitutive and inducible transgenic SBMA mice, together with knock-in mice and human muscle biopsy samples, here we provide the first evidence that mutant AR alters the pathways that represent the foundations for muscle function and structure (triad), mitochondrial respiration, $Ca^{2+}$ dynamics, and the kinetics of muscle contraction.

## Results

### Disruption of muscle contraction dynamics precedes motor dysfunction in SBMA mice

Here, we investigated the pathological processes that occur in skeletal muscle during SBMA as a function of disease progression. A unique feature of SBMA among other neurodegenerative diseases is that SBMA is tightly androgen dependent. In mice, serum androgen levels increase in the perinatal period and rapidly (within 3–4 days after birth) decrease until puberty (around 4 weeks of age), when they rise again[27]. Mice reach sexual maturity between 4 and 7 weeks of age. We recently generated and characterized a transgenic mouse model expressing human AR with an expanded polyQ tract (AR100Q)[5]. These mice recapitulated the main characteristics of the disease, including muscle atrophy and weakness in adulthood (8 weeks of age), progressive body weight loss, and motor dysfunction, followed by signs of denervation and neuromuscular junction degeneration at 12 weeks of age (Fig. 1a and Table 1). Thus, we assigned a color code to distinguish three stages: light green for 3-week-old mice (before the androgen peak); green for 4-week-old mice, which did not yet exhibit the onset of motor dysfunction; magenta for 8-week-old mice, which exhibited disease onset; and black for 12-week-old mice, which were in the late stage of disease. We analyzed nerve conduction velocity by stimulating the sciatic nerve and measuring the compound muscle action potential (CMAP) in innervated fast glycolytic muscles, namely, the tibialis anterior (TA) and gastrocnemius, which are severely affected in the SBMA[5,26]. Consistent with previous findings in SBMA patients[28] and mice[29], 8-week-old SBMA mice showed a dramatic reduction in the CMAP in the TA and gastrocnemius muscles starting at 8 weeks of age (Fig. 1b, Supplementary Fig. 1a). However, we did not find any difference in Hoffmann's reflex (H-reflex), which measures the reflectory reaction of muscles after the electrical stimulation of sensory fibers (Ia afferents) stemming from muscle spindles in their innervating nerves (Supplementary Fig. 1b).

Since the defects in CMAP were consistent with both a neurogenic and a myogenic origin, we measured intrinsic muscle force by direct ex vivo electrical stimulation of the extensor digitorum longus (EDL), a fast glycolytic muscle that shows signs of severe atrophy, and the soleus, a slow-oxidative muscle that is less compromised in SBMA mice[5,26]. Although normal at 4 weeks of age, the maximum specific twitch force ($P_t$) and tetanic force ($P_0$) decreased by approximately 75% and 71%, respectively, at 8 weeks of age in the EDL (Fig. 1c, Supplementary Fig. 1c); no such reduction was observed in the soleus (Fig. 1d). Next, we investigated functional defects in muscle contraction dynamics. We found an increase in fiber contraction and half-relaxation time in the EDL muscle of 4- and 8-week-old AR100Q mice, indicating slower contraction kinetics (Fig. 1e), consistent with previous findings in SBMA patients[30] and mice[15,31]. The slowing of contractile speed in association with muscle fatigue allows the tetanic fusion of force to occur at lower firing frequencies, thus causing a leftward shift of the force–frequency curve. Consistent with this

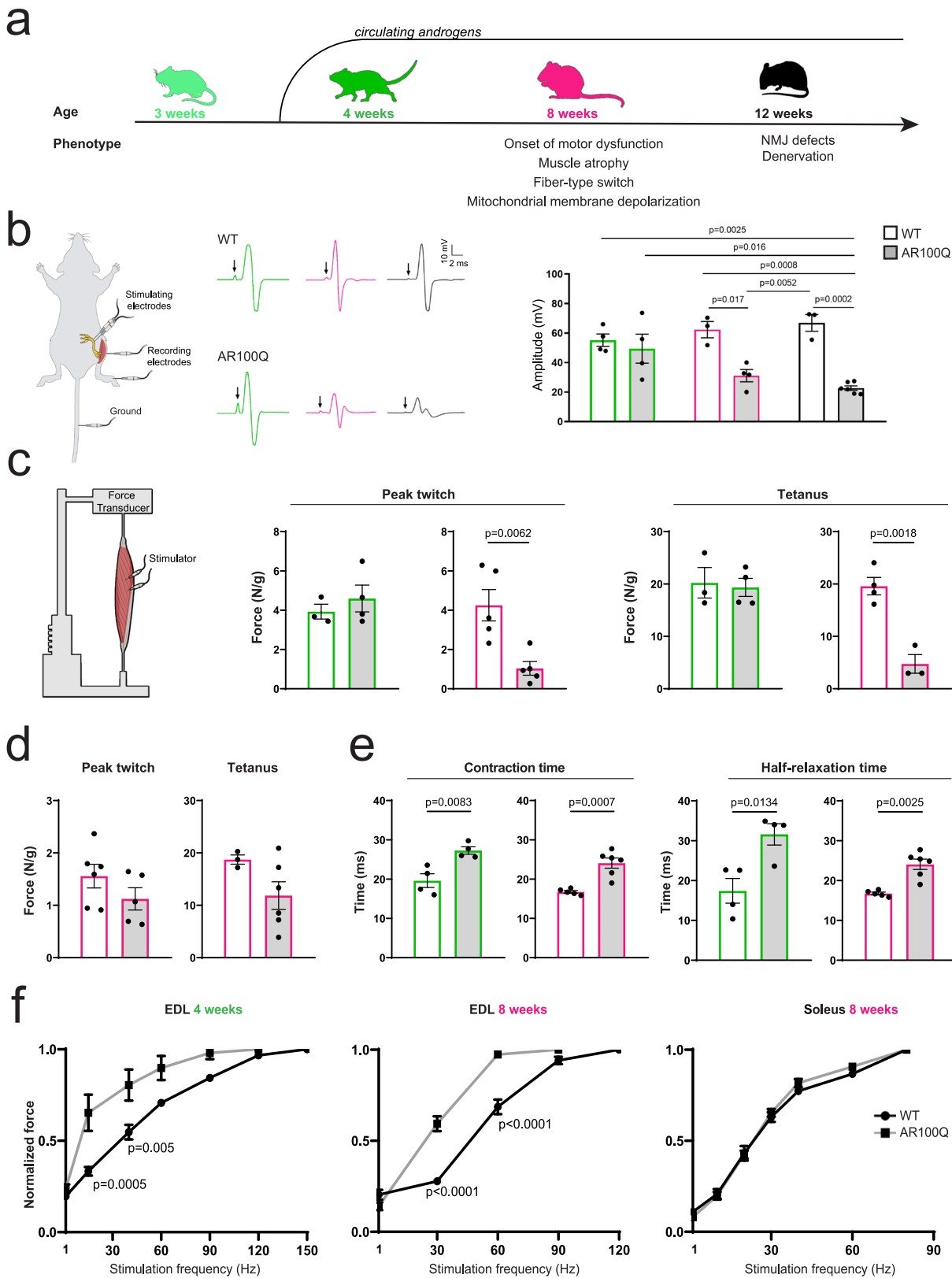

notion, increased relaxation time caused a leftward shift in the normalized force–frequency curve, showing the anticipated fusion of stimuli, with SBMA fibers reaching normalized maximal force in response to a lower frequency of stimulation: at 8 weeks of age, the maximum tension was reached at 60 Hz in the AR100Q EDL muscle, compared to 120 Hz in the WT EDL muscle (Fig. 1f). This phenomenon

was not observed in the soleus. The shift in the normalized force–frequency curve was already established in the EDL at 4 weeks of age (Fig. 1f), which was before any alterations emerged in the production of the maximum twitch force ($P_t$) or the tetanic force ($P_0$) (Fig. 1c). These findings indicate the presence of functional alterations in the contraction dynamics of fast-twitch SBMA muscles. This

**Fig. 1 | Prolonged muscle contraction/relaxation time precedes defects in intrinsic muscle force generation in SBMA mice. a** Scheme of disease progression in AR100Q-transgenic mice. We assigned a color code to distinguish three stages of the disease: green: 4 weeks of age - before the onset of motor dysfunction; magenta: 8 weeks of age - the onset of motor dysfunction; black: 12 weeks of age - late stage of the disease. The phenotype was previously described[5]. **b** CMAP analysis in the TA muscle of 4-week-old (green), 8-week-old (magenta), and 12-week-old (black) WT and AR100Q mice (each dot is the average of 3 measures/muscle; $n = 4$ mice/WT/1M; $n = 4$ mice/AR100Q/1M; $n = 3$ mice/WT/2M; $n = 4$ mice/AR100Q/2 M; $n = 3$ mice/WT/3M;). **c** Generation of intrinsic force in the EDL muscle of 4-week-old (green) and 8-week-old (magenta) WT and AR100Q mice (single twitch: $n = 3$ mice/WT/1 M; $n = 4$ mice/AR100Q/1M; $n = 5$ mice/WT/2M; $n = 5$ mice/AR100Q/2M; tetanus: $n = 3$ mice/WT/1 M; $n = 4$ mice/AR100Q/M; $n = 4$ mice/

WT/2 M; $n = 3$ mice/AR100Q/2M). **d** Generation of intrinsic force in the soleus muscle of 8-week-old (magenta) WT and AR100Q mice (single twitch: $n = 6$ mice/WT/2M; $n = 5$ mice/AR100Q/2 M; tetanus: $n = 3$ mice/WT/2M; $n = 6$ mice/AR100Q/2M). **e** Peak contraction and half-relaxation time in the EDL muscle of 4-week-old (green) and 8-week-old (magenta) WT and AR100Q mice (contraction time: $n = 4$ mice/WT/1 M; $n = 4$ mice/AR100Q/1M; $n = 5$ mice/WT/2 M; $n = 6$ mice/AR100Q/2 M; half-relaxation time: $n = 4$ mice/WT/1 M; $n = 4$ mice/AR100Q/1M; $n = 5$ mice/WT/2 M; $n = 5$ mice/AR100Q/2M). **f** Intrinsic muscle force/frequency in the indicated muscles of WT and AR100Q mice (EDL: $n = 3$ mice/WT/1M; $n = 4$ mice/AR100Q/1M; $n = 5$ mice/WT/2M; $n = 6$ mice/AR100Q/2M; Soleus: = 6 mice/genotype/2M). The graphs show the mean ± SEM; significance was tested with two-way ANOVA followed by Tukey's HSD test (**b, f**) or by the two-tailed Student's *t*-test (**c-e**). Source data are provided as a Source Data file.

## Table 1 | Disease progression in different SBMA mouse models

|  | Presymptomatic | Disease onset | Advanced stage | Reference |
|---|---|---|---|---|
| AR100Q Transgenic mice | 4 weeks of age | 8 weeks of age | 12 weeks of age | [5] |
| AR113Q Knock-in mice | 12 weeks of age | 24 weeks of age | 48 weeks of age | [7] |
| iAR100Q Conditional mice | 4 weeks of age | 7/8 weeks of age | 12 weeks of age | [5] |

aspect of SBMA muscle pathology is not due to decreased neurotransmission and occurs even before the reduction of muscle force, the onset of motor dysfunction, and alterations in fiber-type composition[5].

### Early altered expression of genes involved in muscle contraction in SBMA mice

To elucidate the molecular processes underlying the structural and functional changes responsible for muscle force reduction, we used an unbiased approach and performed RNA sequencing (RNA-seq) analysis in the quadriceps muscle of 4- and 8-week-old AR100Q mice and age-matched WT siblings (Fig. 2a, Supplementary Data 1). We identified 492 differentially expressed genes (DEGs), of which 216 were upregulated and 276 were downregulated ($p \leq 0.05$, and absolute log fold chage $\geq 1$) at 4 weeks of age, and 1155 DEGs, of which 526 were upregulated and 629 were downregulated ($p \leq 0.05$, and absolute log fold chage $\geq 1$) at 8 weeks of age. Gene Ontology (GO) analysis focusing on the category "cellular component" identified early dysregulation of genes related to the terms "sarcomere" and "contractile fiber"; this dysregulation was exacerbated by age (Fig. 2b). GO analysis of the category "biological process" showed early changes (4 weeks of age) in "striated muscle contraction" genes; these changes were exacerbated in association with changes in "SR Ca$^{2+}$ ion transport" at 8 weeks of age (Supplementary Fig. 2a)[26]. GO analysis of the "molecular function" category identified expression changes in gene clusters related to "actin binding" and "serine peptidase activity" at 4 weeks of age, followed by exacerbation of the changes in the "actin binding" cluster and the emergence of expression changes related to voltage-gated ion channels and transmembrane transporter pathways (Supplementary Fig. 2b).

To further corroborate these results, we performed microarray analysis in the quadriceps muscle of knock-in mice expressing a humanized AR with 113 CAG repeats[7]. As a control, we used mice expressing a humanized AR with 21 CAG repeats. We have previously reported that these mice develop motor dysfunction after 20 weeks of age[19,32–34]. To identify the changes in gene expression that preceded the onset of motor dysfunction, we performed our analysis in 12-week-old mice (Table 1). We identified 1233 DEGs, of which 571 were upregulated and 662 were downregulated ($p \leq 0.05$, and absolute log fold chage $\geq 1$) (Supplementary Fig. 3a, b, Supplementary Data 2). GO analysis for

the category "cellular component" confirmed that the pathways enriched with DEGs were "I band", "Z disk", "sarcomere", "myofibril", and "contractile fiber". Similar results were obtained by analyzing a transcriptomic profile previously performed in the quadriceps of 24-week-old knock-in mice (GEO accession number GSE68441) (Supplementary Fig. 4a)[26]. GO analysis for the category "biological process" showed early changes in metabolic pathways, including glucose and lipid metabolism, which is again consistent with the results of previous transcriptomic analysis in older mice (Supplementary Fig. 4b)[26]. Finally, the GO analysis of the category "molecular function" identified several pathways, such as "receptor regulator activity", "extracellular matrix structural components", and "sulfur compound binding" (Supplementary Fig. 4c).

These results highlight the involvement of early changes in the pattern of expression of genes involved in muscle contraction and structure in SBMA mice.

### Altered expression of key ECC genes in SBMA skeletal muscle in a presymptomatic stage

In our transcriptomic analysis *Sln* was the gene most upregulated in 24-week-old knock-in mice[26]. By real-time PCR analysis, we found that the transcript levels of key ECC genes, including *Cacna1s* (which encodes a subunit of DHPR), *Ryr1*, *Atp2a1*, *Casq1* and *Pv*, were significantly decreased in the quadriceps muscles of 8-week-old transgenic SBMA mice, while *Casq2* and *Sln* were upregulated (Fig. 3a). ECC gene expression was normal by 3 weeks of age, when serum androgen levels are low. Importantly, changes in *Atp2a1*, *Casq1*, and *Pv* expression occurred in conjuction with the rise of circulating androgen levels (4 weeks of age) and preceded the presence of intrinsic muscle force defects (8 weeks of age). The expression of the ECC genes was altered in other glycolytic muscles, i.e., the EDL and flexor digitorum brevis (FDB) (Supplementary Fig. 5a, b). These abnormalities in gene expression were attenuated or absent in the soleus muscle (Supplementary Fig. 5c). In the quadriceps muscle, we did not detect any change in the expression of other genes that participate in the ECC process, including SH3 and cysteine-rich domain 3 (*Stac3*), Triadin (*Trdn*), Junctin (*Jctn*), and FK506 binding protein 12 (*Fkbp12*) (Supplementary Fig. 5d). We previously showed that postnatal (week 6) expression of mutant AR suffices to induce key manifestations of the disease, including muscle atrophy, body weight loss, and premature

## a

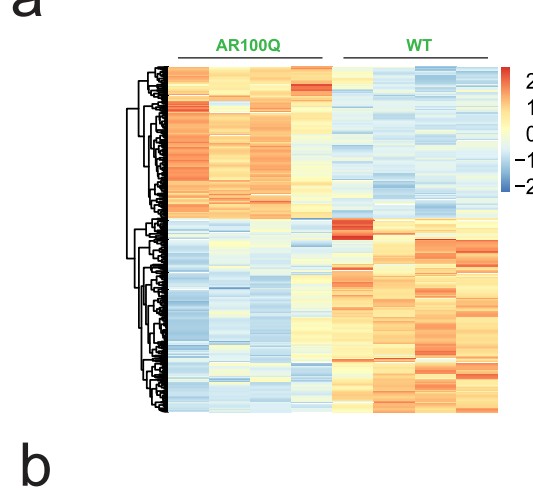

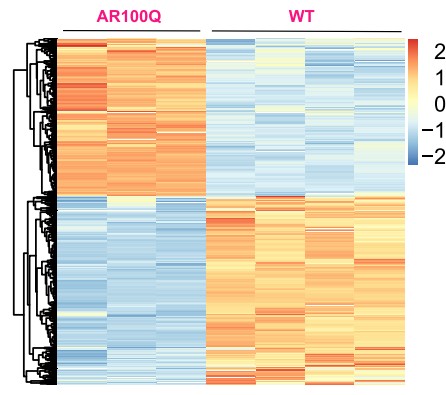

## b

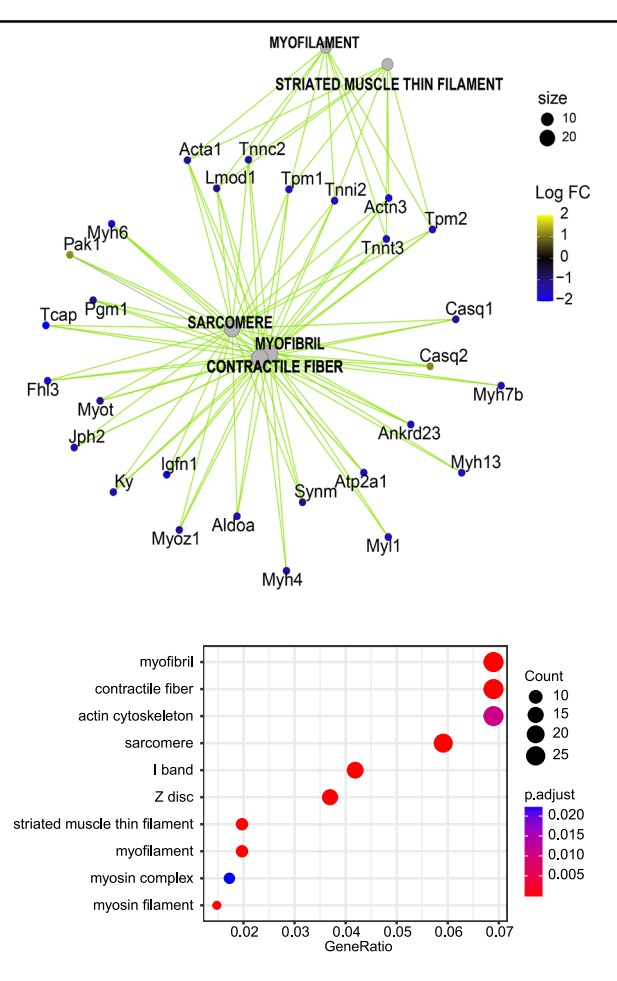

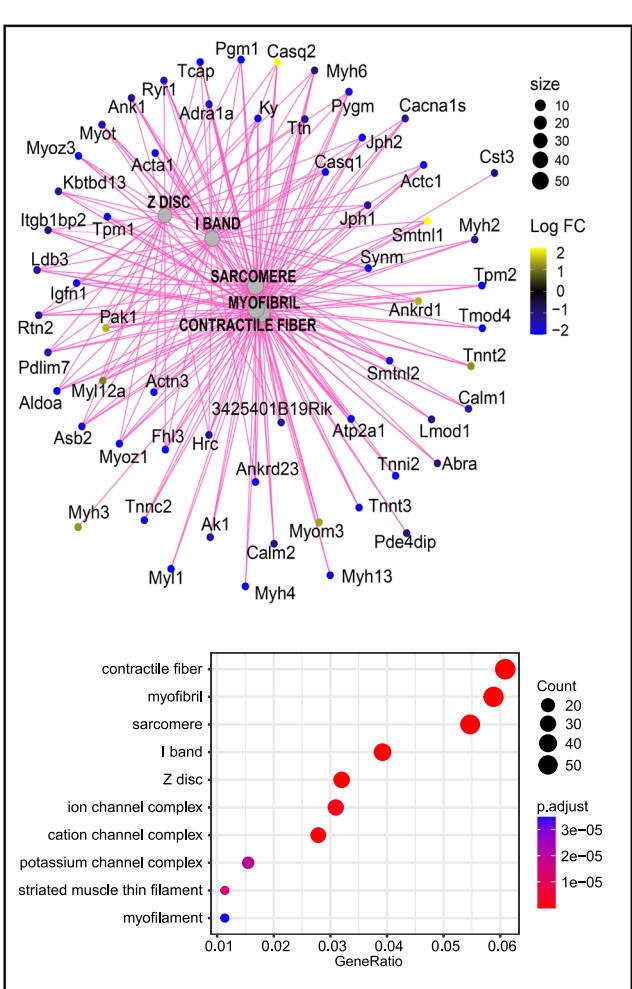

**Fig. 2 | Progressively altered expression of genes involved in sarcomere organization and muscle contraction in SBMA transgenic mice. a** Transcriptomic (RNA-seq) analysis in the quadriceps muscle of 4-week-old (green) and 8-week-old (magenta) AR100Q mice and WT mice ($n = 4$ mice/WT/1M; $n = 4$ mice/AR100Q/1M; $n = 3$ mice/WT/2M; $n = 4$ mice/AR100Q/2M). **b** Gene Ontology analysis of differentially expressed genes in the category "cellular component".

death (Table 1)[5]. By treating inducible AR100Q (iAR100Q) transgenic mice with doxycycline from weeks 6 to 8, we found that the transcript levels of specific ECC genes, namely, *Casq1*, *Pv*, and *Sln*, mirrored those observed in AR100Q mice, indicating that acute induction of polyQ-expanded AR expression in adulthood is sufficient to affect the expression of these ECC genes (Fig. 3b). Importantly, we confirmed our

findings in knock-in SBMA mice, indicating that most changes in gene expression were not the result of AR overexpression (Fig. 3c). By western blot analysis, we verified that the changes in mRNA levels correlated with changes at the protein level in transgenic mice (Fig. 3d). These observations revealed that in SBMA fast-twitch muscles, the expression of ECC genes is altered, and this is an early

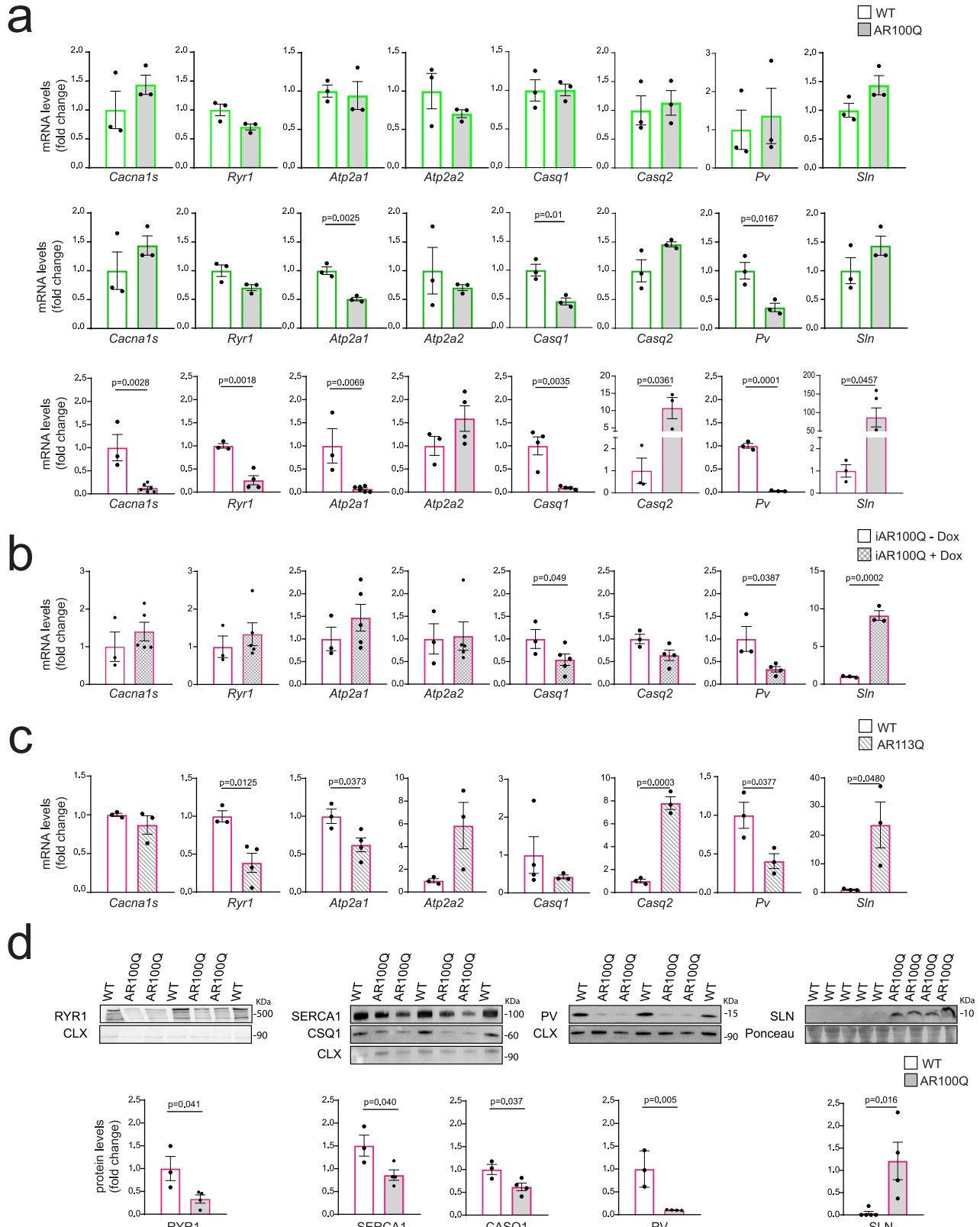

phenomenon that precedes the decrease in intrinsic muscle force generation.

## Defects in myofiber respiration prior to motor dysfunction in SBMA mice

Mitochondria are located close to the triad and are required for ATP production and Ca²⁺ buffering during muscle contraction[35]. The mitochondria in skeletal muscle are depolarized in 8-week-old transgenic AR100Q mice[5] and 24-week-old knock-in AR113Q mice[26] and then are degraded through mitophagy[11]. We thus analyzed the subcellular localization of RYR1, the major RYR isoform expressed in skeletal muscle, starting at 4 weeks of age, an age at which glycolytic fibers do not show a significant reduction in cross-sectional area, and at 8 weeks of age, at which glycolytic fibers show severe

**Fig. 3 | Altered expression of the ECC genes in the muscle of SBMA mice. a–c** RT-PCR analysis of the transcript levels of the indicated ECC genes normalized to *beta-actin* in the quadriceps muscle of **a** 3-week-old (light green, top row), 4-week-old (green, middle row) and 8-week-old (magenta, bottom row) WT and AR100Q mice ($n = 3$ mice/genotype/p20-1M; Cacna1s/Atp2a1: $n = 3$ mice/WT/2M, $n = 6$ mice/AR100Q/2M; Ryr1/Atp2a2: $n = 3$ mice/WT/2M, $n = 4$ mice/AR100Q/2M; Casq1: $n = 4$ mice/WT/2 M, $n = 4$ mice/AR100Q/2M; Casq2/Pv/Sln: $n = 3$ mice/WT/2M, $n = 3$ mice/AR100Q/2M). **b** 8-week-old iAR100Q mice treated with doxycycline (1 g/L), (Cacna1s/Ryr1/Atp2a1/Atp2a2/Casq1: $n = 3$ mice/WT, $n = 5$ mice/iAR100Q; Casq2/Pv: $n = 3$ mice/WT, $n = 4$ mice/iAR100Q; Sln: $n = 3$ mice/WT, $n = 2$ mice/iAR100Q); **c** 24-week-old knock-in AR113Q mice (Cacna1s/Casq2/Pv/Sln: $n = 3$ mice/WT, $n = 3$ mice/AR113Q; Ryr1/Atp2a1/Atp2a1: $n = 3$ mice/WT, $n = 3$ mice/AR113Q; Casq1: $n = 4$ mice/WT, $n = 3$ mice/AR113Q). **d** Western blot analysis of the indicated ECC genes in the quadriceps muscle of 8-week-old WT and AR100Q mice (RYR1/SERCA1/CASQ1/PV: $n = 3$ mice/WT, $n = 4$ mice/AR100Q; SLN: $n = 5$ mice/WT, $n = 4$ mice/AR100Q). ECC genes were detected with specific antibodies, and calnexin (CNX) or Ponceau S were used as a loading control. The quantification (total/CNX, or SLN/Ponceau S) normalized to each control set as 1 is shown at the bottom of each panel. The graphs show the mean ± SEM; significance was tested by the two-tailed Student's t-test. Source data are provided as a Source Data file.

atrophy[5]. By immunofluorescence analysis of transverse sections from the quadriceps of AR100Q mice, we observed areas deprived of RYR1 expression exclusively in fast-twitch fibers at 8 but not 4 weeks of age (Fig. 4a and Supplementary Fig. 6a), a pattern that resembles the muscle pathology of SBMA patients[11]. By Seahorse analysis, we found that the basal and maximal oxygen consumption rates (OCRs) were lower in myofibers isolated from the FDB muscles of 8-week-old SBMA mice than in those obtained from WT mice (Fig. 4b). Importantly, defects in OCR were detected as early as 4 weeks of age, indicating that they precede mitochondrial membrane depolarization and RYR1 deprivation[5]. Interestingly, the basal and maximal OCRs in SBMA myofibers were similar, suggesting that basal respiration is already at the maximum level in these muscles. These defects were not due to altered activity of the respiratory chain complexes (Fig. 4c). Serum lactate levels were normal in AR100Q mice (Supplementary Fig. 6b), and mitochondrial pyruvate carrier 2 (*Mpc2*) and carnitine palmitoyltransferase I (*Cpt1*) transcript levels were both higher in the skeletal muscle of AR100Q mice than in that of WT mice (Supplementary Fig. 6c). These results indicate that dysfunctional myofiber respiration is an early event (4 weeks of age) that precedes mitochondrial membrane depolarization and mitophagy (8 weeks of age)[5].

## Altered Ca²⁺ dynamics and activity-dependent accumulation of Ca²⁺ in SBMA mitochondria

The findings described above prompted us to analyze Ca²⁺ dynamics in SBMA myofibers. Using enzyme-dissociated FDB fibers and the ratiometric Ca²⁺ dye Fura-2, we found that, upon single-twitch stimulation, the kinetics associated with reaching the peak in cytosolic Ca²⁺ levels were prolonged in AR100Q mice, suggesting an altered balance between SR Ca²⁺ release and reuptake (Fig. 5a). Additionally, we observed a significant delay in the decrease in the Fura-2 signal in AR100Q mice, suggesting slow kinetics of myoplasmic Ca²⁺ clearance (Fig. 5b, c). By monitoring cytosolic Ca²⁺ dynamics upon tetanic stimulation, we found that the increased level of cytosolic Ca²⁺ that normally occurs after delivering multiple impulses at high frequency was significantly smaller in AR100Q myofibers than in control myofibers ($\Delta R = 0.08$ vs. 0.12 in WT mice, $p < 0.0001$) (Fig. 5c). Importantly, we also detected slower kinetics of decay time to return to the basal level after tetanic stimulation (67.1 ms in AR100Q vs. 55.4 ms in WT mice) in AR100Q mice (Fig. 5d). Although these changes in cytosolic Ca²⁺ increments did not cause a significant alteration in the cytosolic Ca²⁺ levels in the AR100Q myofibers (Fig. 5e), the impaired balance of cytosolic Ca²⁺ entry and clearance between tetanic stimuli led to a redistribution of cytosolic Ca²⁺ levels (Fig. 5f), which is consistent with the alterations observed in the dynamics of muscle contraction (Fig. 1). Notably, the slower cytosolic Ca²⁺ entry that was observed in the initial phase (first 10 ms of Fig. 5f) led to slower contraction of the myofibers, whereas the prolonged cytosolic Ca²⁺ decay time caused a delay in the time needed for muscle relaxation. We did not observe any change in cytosolic Ca²⁺ levels under basal conditions or in response to caffeine stimulation in 8-week-old mice (Fig. 5g). Using a mitochondrial Ca²⁺-sensitive probe[36], we found a significant increase in the levels of mitochondrial Ca²⁺ accumulation upon caffeine stimulation in 8-week-old AR100Q mice (Fig. 5h) and in 16- and 24-week-old AR113Q knock-in mice (Fig. 5i). These findings show that SBMA mice exhibit impaired Ca²⁺ dynamics in the different subcellular compartments during intense myofiber activity, which is associated with late activity-dependent accumulation in mitochondria.

## Myofiber organization is disrupted in SBMA mice at the late stage of the disease

To establish whether the molecular and functional defects described above were associated with structural changes, we analyzed the organization of the myofiber by immunofluorescence. RYRs are positioned at the junctional SR (J-SR), which is the region of the terminal SR cisternae that faces the t-tubules in triads between longitudinal SR (L-SR) (Fig. 6a). By confocal microscopy and assessments of the fluorescence intensity profile of RYR1 and CASQ, we found that the regular doublet pattern of Ca²⁺-release units in TA myofibers was completely disrupted in 12-week-old SBMA mice (Supplementary Fig. 7). Mitochondria are in close proximity to the SR and form tight association areas between the two organelle membranes, known as mitochondria-associated membranes (MAMs), which face each other. By analysis of translocase of outer mitochondrial membrane 20 (TOM20) and RYR1 expression, we found a pattern of distribution of the mitochondria in the TA bundle fibers of 12-week-old AR100Q mice that was different from that observed in age-matched control mice (Fig. 6b). Similar results were obtained by analyzing the distribution of the expression of actinin, a microfilament protein that links actin filaments to the Z-lines necessary for the coordinated movement of sarcomeres (Supplementary Fig. 8). All these abnormalities were not detected at 4 weeks of age and were mild or absent at 8 weeks of age. Together, these observations reveal a progressive alteration of myofiber organization in the skeletal muscle of SBMA mice.

## Suppression of androgen signaling normalizes mitochondrial respiration, myofiber structural changes, and ECC gene expression

Surgical castration in animal models and pharmacological treatment in patients to reduce serum androgen levels improve the SBMA phenotype[4,6,7,37,38]. Therefore, we asked whether the early muscle pathological processes described above were androgen dependent and reversible. To address these questions, we performed surgical castration between 4 and 5 weeks of age (Fig. 7a). Surgical castration of SBMA mice restored the OCR to normal levels (Fig. 7b) and reduced the changes in ECC gene expression observed in SBMA muscle (Fig. 7c). Amelioration of these early defects was associated with improvements in muscle pathology (Supplementary Fig. 9) and myofiber structure (Fig. 7d, Supplementary Fig. 10), indicating that these pathological processes are androgen dependent and suggesting a causal link between early dysfunctional processes involving the main components of the triads and the structural/functional properties of striated muscles.

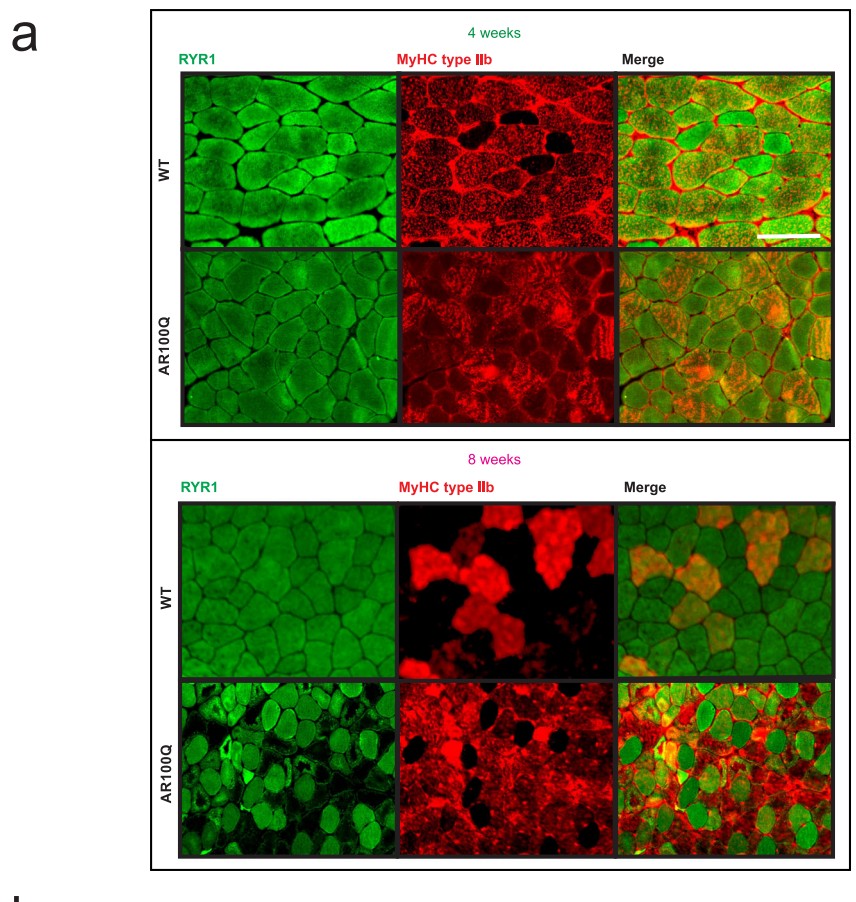

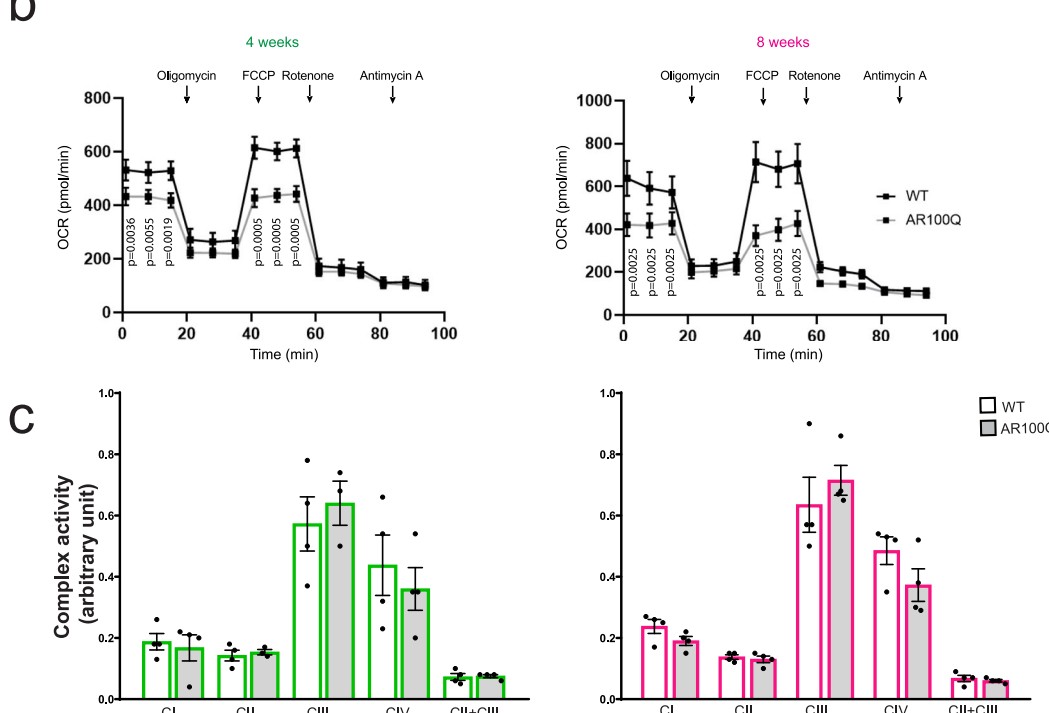

**Fig. 4 | Early defects in mitochondrial respiration in the muscle of SBMA mice.** **a** Immunofluorescence analysis of RYR1 (green) and MyHC type IIb (red) expression in the quadriceps of WT and AR100Q mice (*n* = 3 mice/genotype/age). Representative images are shown. Bar, 40 μm. **b** Analysis of the OCR in FDB-isolated fibers from 4- and 8-week-old WT and AR100Q mice (*n* = 4 mice/genotype/age). Oligomycin 2 μM, carbonyl cyanide-p-trifluoromethoxyphenylhydrazone (FCCP) 0.6 μM, rotenone 1 μM, and antimycin A 1 μM. **c** Mitochondrial complex (C) I, II, III, and IV activity normalized to mitochondrial citrate synthase activity measured in the quadriceps of 4- and 8-week-old WT and AR100Q mice (*n* = 3 mice/genotype/age). The graphs show the mean ± SEM; significance was tested by two-way (**b**) and one-way (**c**) ANOVA followed by Tukey's HSD test. Source data are provided as a Source Data file.

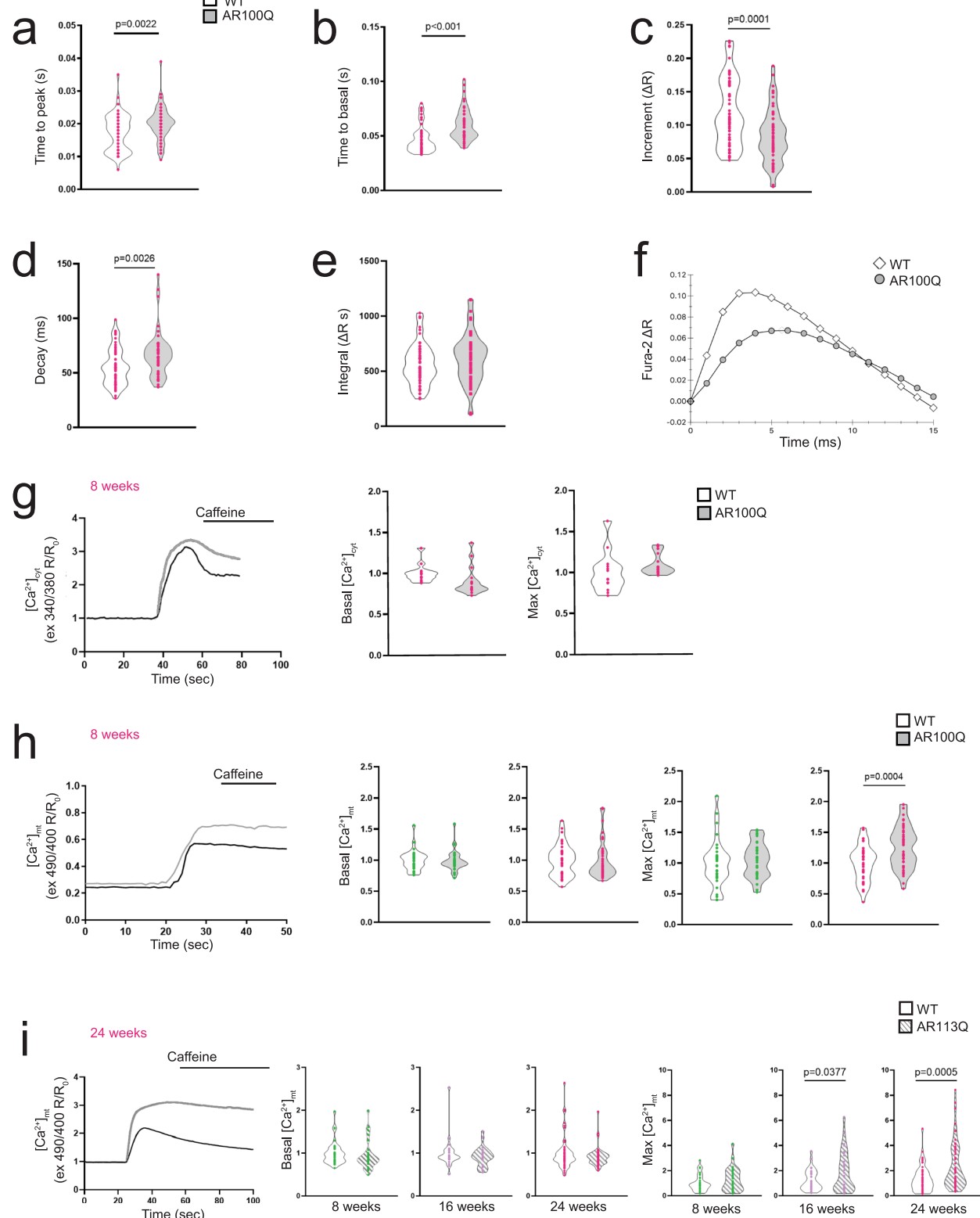

A gene therapy approach for gain-of-function (GOF) diseases that is currently under investigation involves silencing of the diseased gene. This approach has been pursued in SBMA with the use of ASOs in knock-in AR113Q mice[19]. Treatment of knock-in SBMA mice with ASOs to silence the *Ar* gene reduced changes in ECC gene expression (Fig. 7e). These observations indicate that dysregulation of ECC gene expression requires sustained expression of polyQ-expanded AR during disease progression.

**ECC gene expression is altered in the muscle of SBMA patients**
Next, we assessed whether the expression of ECC genes was altered in SBMA patients. By Western blot analysis, we detected significant changes in the protein levels of RYR1, SERCA1, CASQ, PV, and SLN in the muscle of SBMA patients, highlighting the biological relevance of our findings to the disease (Fig. 8a). In conclusion, we provide evidence that defects in myofiber respiration, expression of ECC genes, and contraction dynamics as well as an increase in fatigue are novel early

**Fig. 5 | Stimulus-induced accumulation of Ca²⁺ in mitochondria in SBMA mice.** Analysis of Ca²⁺ levels in response to twitch (**a**, **b**) and tetanic (**c**, **d**) stimulation in FDB-isolated fibers of 8-week-old WT and AR100Q mice (number of myofibers derived from 3 mice/genotype: n = 65 WT and *n* = 66 AR100Q in **a**, **b**, and *n* = 58 WT and *n* = 53 AR100Q in c-d). **e**, **f** Analysis of Ca²⁺ levels in response to tetanic stimulation in fibers isolated from FDB of 8-week-old WT and AR100Q mice (*n* = 3 mice/genotype). The amount of Ca²⁺ accumulated in the cytosol during tetanic stimulation (**a**), calculated as the integral of the FURA_2 ratio in one second, did not show significant differences between WT and AR100Q mice. The representative

ensemble average of the Ca²⁺ transients between consecutive stimuli (f) showed a faster increase as well as a faster decrease in WT than in AR100Q mice. **g**–**i** Resting and peak Ca²⁺ transients in response to caffeine (10 mM) in the cytosol (cyt, **g**) and mitochondria (mt, **h**, **i**) in FDB-isolated myofibers from AR100Q mice (**g**, **h**) and knock-in ARI13Q mice (**i**) and relative control mice (cyt n ≥ 12, mt n ≥ 30 measures in myofibers derived from 3 mice/genotype/age). The graphs show the mean ± SEM; significance was tested by the two-tailed Student's t-test. Source data are provided as a Source Data file.

and reversible components of SBMA muscle pathology and may represent good therapeutic targets for clinical practice (Fig. 8b).

## Discussion

Here, we show that deregulation of ECC gene expression and decreased mitochondrial respiration occur early (before motor dysfunction) in skeletal muscle during SBMA and are followed by mitochondrial Ca²⁺ overload and, later on, by dismounting of sarcomere structure. These findings were detected in skeletal muscle biopsy samples derived from SBMA patients, highlighting their clinical relevance. Importantly, these pathological processes were pharmacologically rescued by a therapeutic intervention that is known to ameliorate the SBMA phenotype.

In SBMA, glycolytic muscles are more severely affected than oxidative muscles[5,26,39]. Consistent with these previous findings, alterations in muscle force and changes in the dynamics of muscle contraction were more prominent in glycolytic muscles than in oxidative muscles. An interesting observation here was that, although muscle force was lower in the SBMA model than in WT mice, its maximum was reached at a lower frequency of stimulation. The biological significance of this finding is that glycolytic muscles reach fatigue faster in SBMA mice than in WT mice. This is important, as patients complain that fatigue is one of the first symptoms to emerge; the onset of fatigue is often used to define the time of disease onset. Trascriptomic analysis performed in the skeletal muscle of early-stage (before motor dysfunction) knock-in and transgenic SBMA mice revealed that the expression of genes encoding sarcomere contractile proteins and proteins involved in the ECC process is dysregulated long before the presence of muscle atrophy, weakness and motor dysfunction. SBMA muscles undergo a progressive fiber-type switch from glycolytic to oxidative during SBMA, which can contribute to changes in gene expression[26]. Indeed, *Atp2a1* and *Casq1* are predominantly expressed in fast-twitch muscles, and their expression is lost in SBMA muscles, whereas *Atp2a2* and *Casq2* are predominantly expressed in slow-twitch muscles, and their expression is spared in SBMA muscles. Notably, glycolytic muscles showed differences in the pattern of changes in ECC gene expression, suggesting that specific subtypes of glycolytic muscles may be more vulnerable than others to the toxicity of polyQ-expanded AR. Several observations support the idea that these alterations are triggered by the interaction of the mutant AR protein with androgens: (i) the induction of polyQ-expanded AR expression in adulthood was sufficient to trigger changes in the expression of *Casq1*, *Pv*, and *Sln* genes, indicating that these alterations are unlikely to be the result of secondary pathological processes occurring during muscle development; (ii) ECC gene expression was normal in 3-week-old SBMA mice, in which serum androgen had not yet reached a plateau; (iii) both the silencing of the *Ar* gene and surgical castration restored the pattern of ECC gene expression back to normal. Thus, altered expression of ECC genes, a phenomenon that occurs early, may be causative or may contribute to the progressive deterioration of SBMA skeletal muscle[5,26,39].

Alteration of the ECC machinery results in a wide range of myopathic conditions, spanning from myotonia to weakness, paralysis, and muscle wasting, which are classified as triadopathies[40,41]. Loss-of-function mutations in the *CACNA1S* and *RYR* genes lead to malignant

hyperthermia and specific forms of congenital myopathy, such as central core disease[42,43], which are characterized by muscle weakness, centrally located cores, and lack of mitochondria and oxidative enzymes with a disorganized contractile apparatus[44]. This pattern of muscle pathology is also observed in SBMA patients[11], and SBMA mice. Decreased levels of DHPR, biochemical alteration of RYR, and DHPR-RYR1 uncoupling correlate with the loss of intrinsic force in aged muscles[45,46]. In SBMA muscles, we found decreased expression of *Cacna1s* and *Ryr1* at the onset of motor dysfunction. Additionally, the expression of other ECC genes, namely, *Atp2a1*, *Casq1*, *Pv*, and *Sln* was altered before the onset of motor dysfunction in SBMA mice. Alteration of SERCA function has also been associated with myotonic dystrophy and hypothyroid myopathy, and decreased activity or abundance levels of this protein results in the selective atrophy of fast-twitch myofibers[47]. Interestingly, myotonia-like symptoms have also been reported in patients with SBMA[48]. In skeletal muscle, SERCA activity is regulated by SLN. SLN overexpression was found to reduce muscle force, slow contraction and relaxation rate and alter Ca²⁺ reuptake into the SR[49], while its ablation had the opposite effect[50]. SLN levels must therefore be tightly regulated in muscles. Indeed, the expression of the SERCA2 and SLN proteins is upregulated in several diseases, such as Duchenne muscular dystrophy (DMD)[51], dysferlinopathies[52], nemaline myopathy[53], and (as shown here) SBMA. A gene therapy approach to decrease SLN levels has been shown to have beneficial effects in DMD mice[54], thus showing that SLN is a potential therapeutic target for muscle-related diseases.

Ca²⁺ homeostasis in muscle relies on the activities of PV and CASQ1/2 that buffer the amount of ions present in the sarcoplasm and inside the SR. Patients with *CASQ1* mutations show a late onset followed by slow progression, and symptoms include elevated serum creatine kinase levels, fibrofatty substitution, exercise intolerance, mild age-dependent muscle weakness and atrophy, and the presence of vacuoles reactive to SR Ca²⁺-regulating proteins[55]. Reduced PV expression was reported in arrested development of righting response (*adr*) mice, an animal model of ADR myotonia characterized by abnormally prolonged muscle contraction and half-relaxation time[56]. *CASQ1* ablation was found to alter muscle contractility and increase susceptibility to spontaneous mortality and heat/anesthesia-induced malignant hyperthermia-like episodes[57,58]. Interestingly, the incidence of lethal hyperthermia is higher in male mice than in female *Casq1* null mice, suggesting a pathogenic link between the ECC process, sex steroid hormones, and their receptors[59]. *PV* ablation resulted in a fast-to-slow fiber-type switch[60] and stimulus-dependent accumulation of Ca²⁺ into mitochondria[61]. Several of these symptoms were found in patients with SBMA[14,62–64]. Hovever, the SBMA phenotype is unlikely to be caused solely by the loss of *PV*, as its knock-out in mice results in fiber hypertrophy[61]. Rather, our results support the idea that SBMA muscle atrophy results from the altered expression of several fundamental genes involved in ECC and muscle contraction.

Each cycle of skeletal muscle contraction and relaxation is based on two factors: ATP availability and Ca²⁺ transients. Skeletal muscle mitochondria are located in close proximity to Ca²⁺ release units and supply the myofibers with ATP. Under physiological conditions, Ca²⁺ enters mitochondria through the voltage-dependent anion channel VDAC and reaches the mitochondrial matrix through the

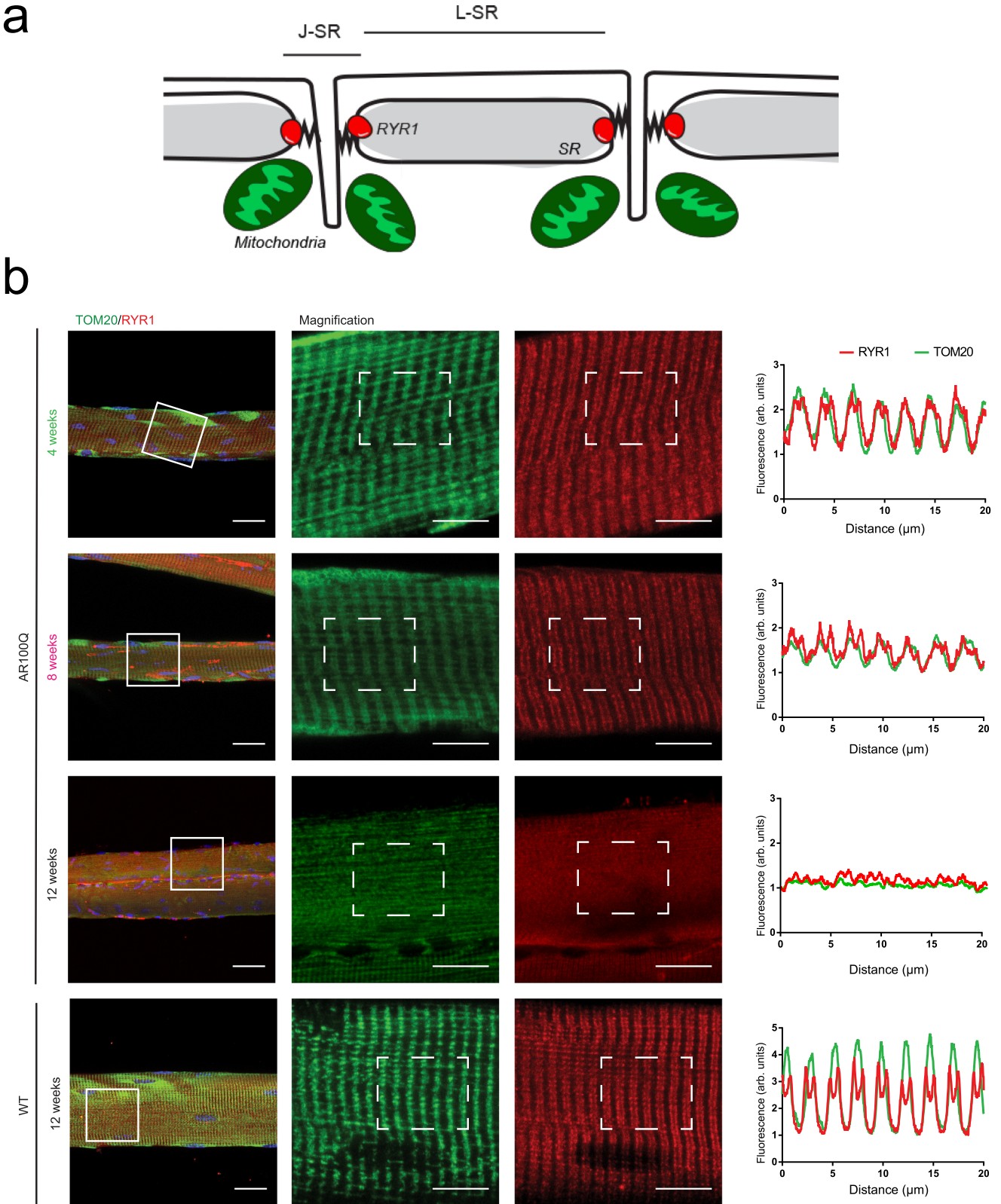

**Fig. 6 | Disrupted mitochondrial organization at the Ca²⁺-release units in the skeletal muscle of SBMA mice. a** Schematic representation of *triads* and *mitochondria* in skeletal *muscle* fibers. **b** Immunofluorescence analysis of RYR1 (red) and TOM20 (green) expression in fibers isolated from the TA muscle of WT and AR100Q mice (*n* = 3 mice/genotype/age). Representative images are shown. Bar = 25 μm, magnification bar = 10 μm. The graphs show fluorescence quantification (described in the "Methods" section).

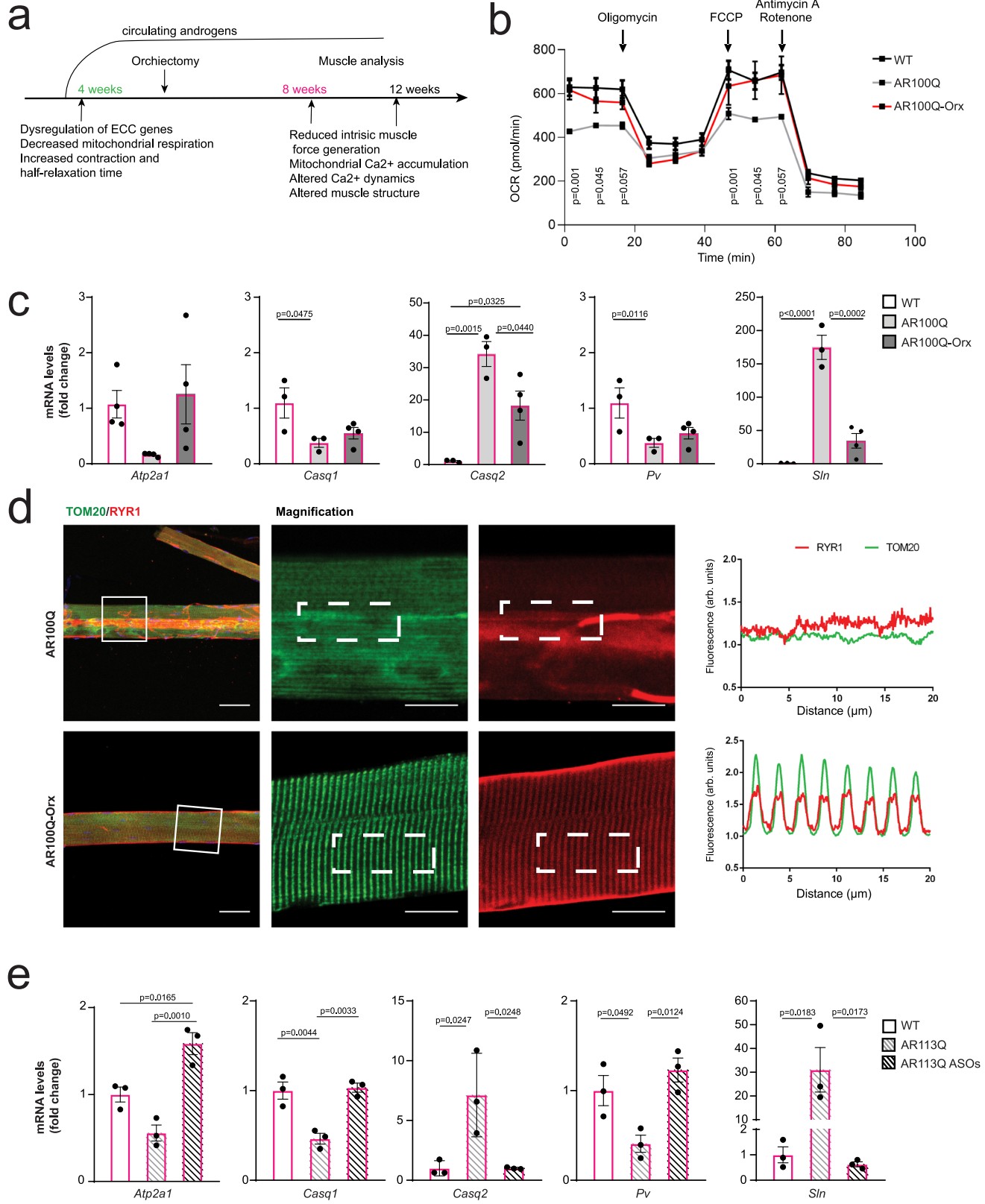

mitochondrial $Ca^{2+}$ uniporter (MCU)[65,66]. $Ca^{2+}$ is then rapidly extruded through the $Na^+/Ca^{2+}$ (NCLX) and $H^+/Ca^{2+}$ (mHCX) exchangers. An unprecedented key finding here is that SBMA muscle is characterized by altered $Ca^{2+}$ homeostasis. SBMA myofibers showed slower release of $Ca^{2+}$ in the cytosol after stimulation and slower kinetics of myoplasmic $Ca^{2+}$ clearance, a necessary step for muscle relaxation. These changes may be at least in part responsible for the longer time needed for muscle contraction and the slower relaxation of the fast SBMA muscles. We detected an accumulation of mitochondrial $Ca^{2+}$ in response to RYR1 stimulation by caffeine. An enhanced accumulation of $Ca^{2+}$ in mitochondria may result from excessive $Ca^{2+}$ flow through the MCU or reduced $Ca^{2+}$ flow through the ion exchanger systems. It is possible

**Fig. 7 | Surgical castration and silencing of polyQ-expanded AR in adulthood normalizes ECC gene expression, mitochondrial respiration, and sarcomere organization. a** Scheme of experiment: Orchiectomy was performed in 4–5-week-old AR100Q mice. **b** OCR analysis in FDB-isolated fibers of sham-operated and orchiectomized (Orx) 8-week-old WT and AR100Q mice (*n* = 4 mice/genotype). **c** RT–PCR analysis of the transcript levels of the indicated genes normalized to *beta-actin* transcript levels in the EDL muscle of sham-operated and AR100Q-Orx 8-week-old mice (Atp2a1: *n* = 4 mice/genotype, Casq1/Casq2/Pv/Sln: *n* = 3 mice/WT, *n* = 3 mice/AR100Q *n* = 4 mice/AR100Q-Orx). **d** Immunofluorescence analysis of RYR1 (red) and TOM20 (green) expression in fibers isolated from the TA muscle of

sham-operated and AR100Q-Orx 12-week-old mice (*n* = 3 mice/genotype). Representative images are shown. Bar = 25 μm, magnification bar = 10 μm. **e** RT–PCR analysis of the transcript levels of the indicated genes normalized to *beta-actin* transcript levels in the quadriceps muscle of 26-week-old AR113Q mice treated with either vehicle or *Ar*-targeting ASOs (*n* = 3 mice/genotype). The graphs show the mean ± SEM; significance was tested by two-way (**b**) and one-way (**c**, **e**) ANOVA followed by Tukey HSD test. In panel (**b**), *p* values correspond to comparison between sham-operated AR100Q and AR100Q-Orx mice. There was no significant difference between AR100Q-Orx and WT mice. Source data are provided as a Source Data file.

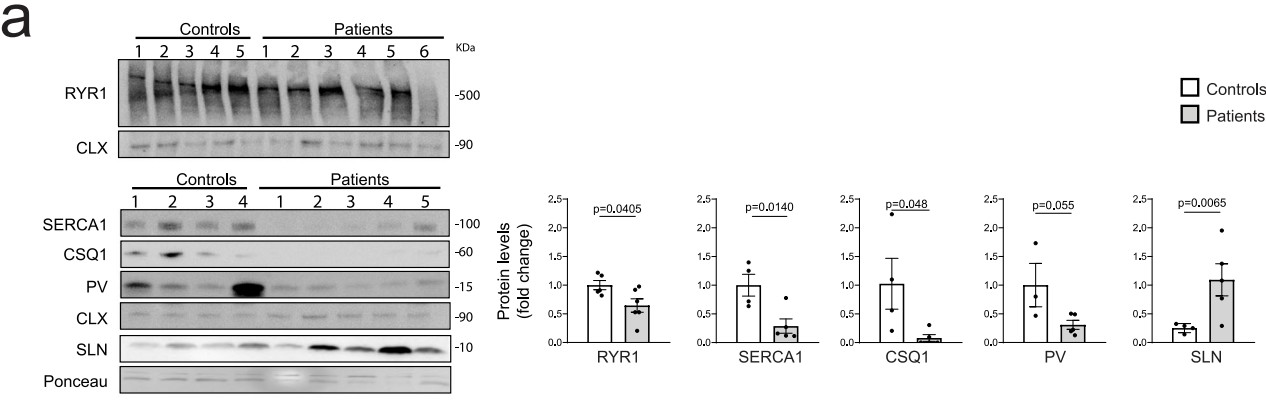

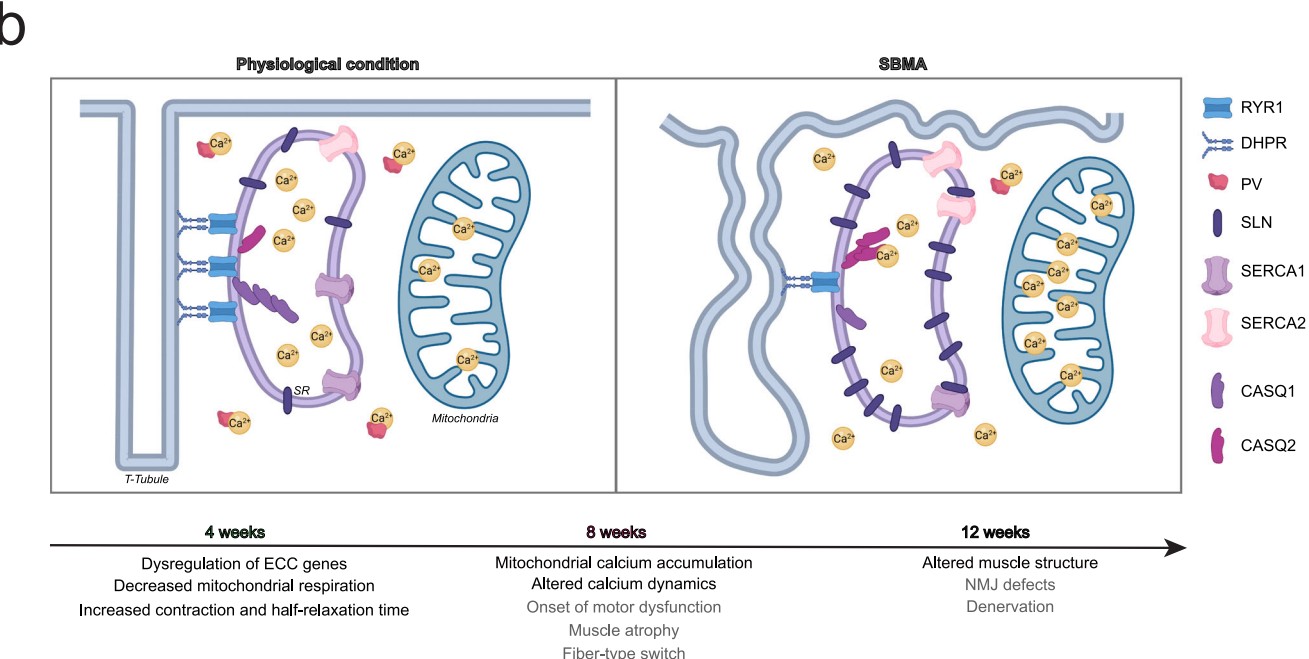

**Fig. 8 | Altered excitation–contraction coupling (ECC) gene expression in the muscle of SBMA patients. a** Western blotting analysis of ECC proteins in skeletal muscle biopsies derived from control subjects and SBMA patients (RYR1: *n* = 5 controls, *n* = 6 patients, CASQ/SERCA1: *n* = 4 controls, *n* = 5 patients, PV: *n* = 3 controls, *n* = 5 patients, SLN: *n* = 4 controls, *n* = 4 patients). ECC proteins were detected with specific antibodies, and calnexin (CNX) or Ponceau S was used as a loading control. The quantification (total/CNX, or SLN/Ponceau S) normalized to each control set as 1 is shown at the bottom of each panel. The graphs show the mean ± SEM; significance was tested by the two-tailed Student's t-test. Source data

are provided as a Source Data file. **b** Working model: In physiological conditions, PV in the cytosol and CASQ1/2 in the SR bind $Ca^{2+}$. Sarcolemma depolarization results in a conformational change of DHPR, which in turn leads to $Ca^{2+}$ efflux through the RYR1 from the SR to the sarcoplasm and into mitochondria. $Ca^{2+}$ then is pumped back to the SR by SERCA1/2, whose function is negatively controlled by SLN. In SBMA-affected muscles, the expression of ECC proteins is altered and $Ca^{2+}$ accumulates into mitochondria during muscle contraction as a function of disease progression. Age refers to AR100Q mice. In gray, pathological processes previously characterized; in black major findings shown here.

that dysfunction of these systems causes mitochondrial $Ca^{2+}$ overload in the SBMA muscle. Under physiological conditions, mitochondrial $Ca^{2+}$ controls oxidative phosphorylation by activating three rate-limiting enzymes of the Krebs cycle, and it controls the respiratory chain and ATP production by stimulating the function of both the ADP transporter and complex V, thus linking the $Ca^{2+}$ transients produced during muscle contraction to ATP production through aerobic respiration[24,67,68]. Due to its important regulatory role, the level of the $Ca^{2+}$ matrix must be tightly regulated. The relevance of the maintenance of $Ca^{2+}$ homeostasis is underscored by pathological conditions characterized by mitochondrial $Ca^{2+}$ overload, such as ischemia–reperfusion of the heart, metabolic syndrome and diabetes and excitotoxicity in neurodegenerative diseases. Mitochondrial $Ca^{2+}$ overload triggers the opening of the permeability transition pore (PTP)[69], which in turn leads to dramatic changes in mitochondrial membrane potential, mitochondrial swelling, hampered ATP production, and activation of death pathways and autophagy, which are all aspects previously described in SBMA muscle[11,26].

A key observation here is that SBMA-affected mitochondria show early defects in respiration. Myofiber respiration is altered likely as a result of multiple factors. We did not detect any defects in circulating lactate levels, which are increased in conditions when oxidative phosphorylation is impaired, while the levels of the mitochondrial pyruvate carriers were enhanced in SBMA-affected muscle. Defects in myofiber respiration did not result from altered respiratory chain complex activity. Furthermore, the activity of complex II + III was normal, which excluded the possibility of deficits in Coenzyme Q. Factors that alter mitochondrial respiration, such as defects in the availability of substrates for glycolysis or the tricarboxylic acid cycle, may be involved upstream. Anabolic pathways, such as the one-carbon cycle, or other biochemical pathways providing precursors for important metabolic processes remain an unexplored aspect of the skeletal muscle pathogenesis of SBMA.

The fact that alterations of ECC gene expression and mitochondrial respiration are evident by 4 weeks of age implies that these pathological processes occur when androgen levels rise and are concomitant with sexual maturity. Indeed, ECC gene expression is normal at 3 weeks of age, when serum androgen levels are very low and sometimes barely detectable[27]. The effect of surgical castration or *Ar* silencing on early and subsequent pathological processes highlights the androgen-dependent nature of all these defects. Thus, early events rapidly occur when androgen levels rise, suggesting that the effect of androgen deprivation treatment in patients may be increased if it is performed early, before the onset of motor dysfunction.

## Methods

### Animals

Our research complies with all relevant ethical regulations. Animal care protocols conform with the appropriate national legislation (art. 31, D.lgs. 26/2014) and guidelines of the Council of the European Communities (2010/63/UE). This study was approved by local ethics committees (University of Trento, University of Padova, Italian Ministry of Health, and the University of Michigan). Mice were housed in filtered cages in a temperature-controlled room with a 12/12-h light/dark cycle with *ad libitum* access to water and food. The mice were pathogen free according to the FELASA list (FELASA 2014). Animals were housed in a single ventilated cage (Tecniplast Green Line Sealsafe PLUS Mouse) with autoclaved commercial soil bedding, food and enrichment. Mice were fed with a certified rodent diet (SDS VRF1 (P)). Mice were monitored on a daily basis by specialized operators and by the designated veterinary. The colonies were monitored by a sentinel program. Mice were euthanized by either administration of carbon dioxide or by the mix of Alfaxalone (60 mg/kg) and Xylazine (10 mg/kg) and were genotyped by PCR on tail DNA using REDExtract-N-Amp Tissue PCR kit

(Sigma-Aldrich, St. Louis, MO, USA) according to the manufacturer's instructions. Genotyping was performed by PCR on ear samples, and primer sequences were as follows: Forward 5′-CTTCTG GCGTGTG ACCGGCG, reverse 5′-TGAGCTTGGCTGAATCTTCC for AR100Q mice (AR100Q); forward 5′-CGTATGTCGAGGTAGGCGTG, reverse 5′-TGAG CTTGGCTGAATCTTCC (iAR100Q); forward: 5′-CCAGAATCTGTTCCA GAGCGTG-3′, reverse: 5′-TGTTCCCCTGGACTCAGATG-3′. (AR113Q)[5,7]. Transgenic lines were backcrossed to the C57Bl6J background for more than 10 generations before subsequent analysis of phenotype and pathology. rtTA mice were purchased from The Jackson Laboratory (Stock n.: 003273). iAR100Q/rtTA and controls mice were treated with doxycycline (Sigma-Aldrich) in drinking water containing 5% sucrose (Sigma-Aldrich S8501). AR113Q male mice and WT littermates were treated for 8 weeks with ASOs. Subcutaneous administration of 50 mg/kg body weight of AR-targeted ASO or control ASO was performed once a week from 6 to 14 weeks of age[19,39]. Orchiectomy (Orx) was performed between 4 and 5 weeks of age. The testes were removed (AR100Q-Orx) or left intact (AR100Q-Sham), and the incision was closed with absorbable sutures.

### Human samples

Deanonymized control and patient biopsy samples were obtained from the Neuromuscular Bank of Tissues and DNA Samples, Telethon Network of Genetic Biobanks, and of the EuroBioBank Network, and Orthopedics and Orthopedic Oncology (University-Hospital of Padua). All muscle biopsies were taken for diagnostic purposes after written informed consent was obtained from each patient according to the Helsinki Declaration. This study was approved by the Ethical committee for Clinical Practice of the Azienda Ospedale Università of Padova. All patients who underwent muscle biopsy were clinically affected and showed weakness and/or muscle atrophy. Information on the age, muscle, clinical status of controls, and Q tract length of SBMA patients is provided in Supplementary Table 1.

### Biochemical analysis

Frozen tissues were pulverized using a mortar and pestle on dry ice and then homogenized in RIPA buffer [150 mM NaCl, 6 mM $Na_2HPO_4$, 4 mM $NaH_2PO_4$, 2 mM ethylenediaminetetraacetic acid (EDTA) pH 8, 1% Na-deoxycholate, 0.5% Triton X-100, 2% sodium dodecyl sulfate (SDS)] and protease inhibitor cocktail (Sigma). Muscle lysates were then sonicated and centrifuged at $12,000 \times g$ for 15 min at room temperature (RT). The protein concentration was measured using the bicinchoninic acid (BCA) assay method. For Western blotting, equal amounts of protein extracts from tissues or cell lysates were boiled in 5X sample buffer (62.5 mM Tris-HCl, pH 6.8, 2% SDS, 25% glycerol, 0.05% bromophenol blue, 5% β-mercaptoethanol). Proteins were separated in 7.5% Tris-HCl SDS polyacrylamide gel electrophoresis (SDS–PAGE) for AR, 3–8% NuPAGE Tris-acetate gels for RYR1, and 4–12% NuPAGE (Thermo Scientific) BisTris gels were used for the other ECC proteins. The gels were blotted overnight on 0.45 μm nitrocellulose membranes (Bio-Rad, 162-0115). The following primary antibodies were used: calnexin (ADI-SPA-860, 1:5000), RYR1 (# MA3-925, 1:2000), CASQ (VIIID12; 1:3000), SERCA1 (VE121G9, 1:50,000), SERCA2 (N19; 1:1000), PV (ab11427, 1:50,000), and SLN[54]. Protein signals were detected using the Li-Cor Odyssey infrared imaging system or the Alliance Q9 Mini Chemidoc system (Uvitec, Cambridge) with appropriate secondary antibodies. Quantifications were performed using ImageJ 1.45 software. For lactate measurement, blood was collected from the tails and lactate levels were measured using the Stat Strip Xpress measuring system following the manufacturer's instructions (Nova Biomedical). For mitochondrial complex activity, total muscle lysates were extracted and the enzymatic activities of respiratory chain complexes I–IV were assayed[26].

## Immunofluorescence and microscopy

Muscles were isolated from mice that were the indicated age and genotype and immediately fixed in 4% paraformaldehyde (PFA) for 15 min at RT. Skeletal muscles were further dissected into muscle bundles of approximately 20 myofibers each. The samples were quenched in 50 mM $NH_4Cl$ for 30 min at RT and then saturated for 2 h in blocking solution [15% vol/vol goat serum, 2% wt/vol bovine serum albumin (BSA), 0.25% wt/vol gelatin, and 0.2% wt/vol glycine in phosphate-buffered saline (PBS) containing 0.5% Triton X-100]. Incubation with primary antibodies against TOM20 (sc-11415, 1:200), RYR1 (MA3-925, 1:2000), CASQ (VIIID12; 1:3000), AR (H280; 1:1000), and ACTININ (mouse monoclonal 1:5000, Sigma–Aldrich) was carried out for at least 48 h in blocking solution. The muscles were then thoroughly washed and incubated with a secondary antibody conjugated with Alexa-555 or Alexa-488 diluted in blocking solution. After this incubation time, we added 4,6 diamidino2-phenylindole (DAPI) solution (D3571, Invitrogen) for 5 min. Images were collected with a Leica SP5 confocal microscope (Leica Microsystems, Wetzlar, Germany) equipped with an objective of 20X HC PL APO 20X/0.75 CS2. The laser excitation line, power intensity, and emission range were chosen according to each fluorophore in different samples to minimize bleed-through. Images were analyzed with the ImageJ program using the 'plot profile' function on a determined area of the sample. For the transverse sections, tissues were flash frozen in isopentane precooled in liquid nitrogen and embedded in an optimal cutting temperature (OCT) compound (Tissue Tek, Sakura, Mestre, Italy). Cross sections (10 µm thick) were cut with a cryostat (CM1850 UV, Leica Microsystems, Wetzlar, Germany) and processed for the analysis of BF-F3 (MyHC-IIb; 1:300), AR (H280, 1:200), RYR1 (MA3-925, 1:200), and WGA (1:500, W11261) expression. The images were captured with a Leica DFC300-FX digital charge-coupled device camera using Leica DC Viewer software and morphometric analyses were performed using ImageJ.

## Quantitative real-time PCR analysis

Total RNA was extracted with TRIzol (Thermo Fisher Scientific), and RNA was reverse transcribed using iScript Reverse Transcription Supermix (1708841 Bio-Rad) following the manufacturer's instructions. Gene expression was measured by RT-qPCR using the SsoAdvanced Universal Sybr green supermix (1725274 Bio-Rad) and the C1000 Touch Thermal Cycler-CFX96 Real-Time System (Bio-Rad). Gene expression was normalized to actin expression levels. The complete list of materials and primer sequences are provided in Supplementary Table 2.

## Single-fiber electrical stimulation

Single digitorum brevis (FDB) fibers were isolated and transfected as previously described[70]. The transients of the fura-2 ratio R were recorded during 2 seconds of stimulations at 60 Hz, with a sampling rate of 1 ms, together with the excitation signals. The data analysis was automated with an ad hoc MATLAB (R) script. After interpolation of the raw data using the built-in interp1 function, the following information was extracted. Basal levels were defined as the average of R in a 500 ms time window before the first impulse of the 60-Hz train. Since different cells required a slightly different number of stimuli to reach steady-state R, we defined the first peak as the maximum value within the first five impulses. The amount of $Ca^{2+}$ released into the cytosolic space was estimated through the integral of R within the first 100 stimuli (approximately 1.6 s). For the transient decay after the last stimulus, R was fitted with a single exponential function in a time window of 600 ms using the built-in Curve Fitting toolbox, and the inverse of the rate of decay (the time constant) is reported in the text. Even if the noise in each trace between two stimuli was too high to provide information, the recording of the stimuli in time in our setup still allowed us to align each stimulus at its onset to obtain the ensemble average. The differences in increments of 61 of the ensemble averages of each cell showed statistical significance between the groups, and the different behaviors can be observed with the mean value of the ensemble averages for each group.

## Cytosolic and mitochondrial $Ca^{+2}$ measurements

For cytosolic $Ca^{+2}$ measurements, the FDB muscles were digested in collagenase A (4 mg/ml) (Roche) dissolved in 1 mL of Tyrode salt solution (pH 7.4) containing 10% FBS. After 1 h on ice and 40 min at 37 °C, the single fibers were mechanically dissociated and plated on glass coverslips coated with laminin in Dulbecco's Modified Eagle Medium (DMEM) with HEPES supplemented with 10% FBS, 1% penicillin (100 U/mL), and 1% streptomycin (100 µg/mL). The day after, the fibers were loaded with 2 µM Fura-2/AM (F1221, Thermo Fisher Scientific) diluted in Krebs Ringer-modified buffer (135 mM NaCl, 5 mM KCl, 1 mM $MgCl_2$, 20 mM HEPES, 1 mM $MgSO_4$, 0.4 mM $KH_2PO_4$, 1 mM $CaCl_2$, 5.5 mM glucose, pH 7.4) containing 0.02% pluronic acid for 20 min at 37 °C and then washed with Krebs Ringer-modified buffer in the presence of 75 µM BTS (1576-37-0, Merck) to prevent fiber contraction. $Ca^{2+}$ release was induced with 40 mM caffeine (C0750, Merck). $Ca^{2+}$ signals were recorded using a Zeiss Axiovert 200 microscope equipped with a 40 × /1.3 N.A. PlanFluor objective. Excitation was performed with a DeltaRAM V high-speed monochromator (Photon Technology International) equipped with a 75 W xenon arc lamp. Images were captured with a high-sensitivity Evolve 512 Delta EMCCD (Photometrics). Data were collected by alternatively exciting the fluorophore at 340 and 380 nm and fluorescence emission was recorded through a 515/30 nm bandpass filter (Semrock). Acquisition was performed at binning 1 with 200 EM gain. Data were analyzed by ImageJ after background extraction. The mean fluorescence was subtracted from the background and R340/400 was then calculated. Data are presented as $R$ or $R/R_0$. For mitochondrial $Ca^{+2}$ measurements, FDB fibers were enzymatically dissociated and plated as described above. The fibers were loaded with 2 µM mt-fura-2[36] and processed as described above. $Ca^{2+}$ release was induced with 40 mM caffeine (C0750, Merck). Images were collected by alternatively exciting the fluorophore at 340 and 400 nm.

## Compound motor action potential (CMAP) and Hoffman's reflex (H-reflex) recording

Mice were anesthetized with xylazine (48 mg/kg) and Zoletil (16 mg/kg) via intraperitoneal injection. The sciatic nerve was exposed at the sciatic notch and a small piece of parafilm was inserted under the nerve, which was kept moist with a drop of PBS. Using a mechanical micromanipulator (MM33, FST, Germany) a pair of stimulating needle electrodes (Grass, USA) were then advanced until they gently touched the exposed sciatic nerve, approximately 0.5 mm below the sciatic notch. We used two needle electrodes for electromyography recording (Grass, USA): the recording electrode was inserted halfway into the selected muscle (gastrocnemius or tibialis for CMAP and FDB for the H-reflex) while the ground electrode was inserted into the distal tendon of the muscle. CMAPs were recorded during supramaximal stimulation of the sciatic nerve (0.4 ms duration, and 0.5 Hz frequency) using a stimulator (S88, Grass, USA) through a stimulus isolation unit (SIU5, Grass, USA) in capacitive coupling mode. Conversely, for the H-reflex we used a lower stimulus intensity (about 1/3 of supramaximal stimulation)[71]. We amplified electromyographic signals with an extracellular GrassP6 amplifier (Grass, USA) and digitized them on a PC using an A/D interface (BNC-2110, NI, USA). Data were recorded with WinEDR software (Strathclyde University, UK) and analyzed offline using pClamp10 software (Axon, USA). The peak-to-peak amplitude was normalized using a 1 V calibration and was compared with the WT amplitude at the same age. Each point is the average of at least 3 CMAPs from a 1-minute registration obtained from each mouse that was analyzed.

### Ex vivo electrical stimulation of muscles

The EDL and soleus were collected and mounted between a force transducer (KG Scientific Instruments, Heidelberg, Germany) and a shaft controlled by a micromanipulator. The muscles were immersed in a chamber in which oxygenated Krebs solution was continuously circulated at 25 °C, and the length of the strips was adjusted until the production of maximal force. The electrical stimulation was carried out using a GRASS S44 module. The stimulation current was transmitted to the bath using platinum electrodes, and the stimulation was performed at 30 V and increasing pulse frequency (ranging from 1 Hz to 150 Hz). Values were normalized to the maximum force of each muscle. The single pulse duration was set to 1 ms, while a single train duration was set to 300 ms. The cross-sectional area was measured from the optimal length and muscle weight.

### Measurements of the oxygen consumption rate

FDB muscles were digested in collagenase A (4 mg/mL) (Roche) dissolved in Tyrode's salt solution (pH 7.4) (Sigma-Aldrich) containing 10% FBS (Thermo Fisher Scientific). Single fibers were isolated; seeded in laminin-coated XF24 microplate wells and cultured in DMEM (D5030 Sigma–Aldrich) supplemented with 1 mM Na pyruvate, 5 mM glucose, 33 mM NaCl, 25 mM HEPES, and 1 mM L-glutamine. Fibers were kept for 2 h in culture at 37 °C under a 5% $CO_2$ atmosphere. The rate of oxygen consumption was evaluated in real time with the Seahorse XF24 Extracellular Flux Analyzer (Agilent). Fibers were plated as reported above. We performed a titration with the uncoupler carbonyl cyanide-p-trifluoromethoxyphenylhydrazone (FCCP). To utilize the FCCP concentration (0.6 μM) that increased the OCR maximum, we loaded the fibers with 2 μM calcein for 30 min, and we normalized the results for the fluorescence levels of calcein expression (Sigma-Aldrich). We measured fluorescence using a PerkinElmer EnVision plate reader in well scan mode using a 480/20 nm filter for excitation and a 535/20 nm filter for emission.

### Microarray and RNA-seq transcriptome analysis

The quality of the RNA was analyzed by microfluidic gel electrophoresis on an RNA 6000 Nanochip using an Agilent 2100 Bioanalyzer (Agilent Technologies Inc., USA). Cyanine-3 (Cy3)-labeled cRNA was prepared from 0.2 μg RNA using the One-Color Low Input Quick Amp Labeling Kit (p/n 5190-2331, Agilent Technologies) according to the manufacturer's instructions (Agilent Technologies, USA), followed by RNeasy column purification (QIAGEN, Valencia, CA). Dye incorporation and cRNA yield were analyzed with a NanoDrop ND-1000 Spectrophotometer (Nano-Drop Technologies, USA), and 1.65 μg of Cy3-labeled cRNA was fragmented at 60 °C for 30 min in a reaction volume of 55 μL containing 1× fragmentation buffer and 2x GE (Gene Expression) Blocking Agent (Gene Expression Hybridization Kit, p/n 5188-5242, Agilent Technologies) according to the manufacturer's instructions. At the end of the fragmentation reaction, 55 μL of 2× Hi-RPM hybridization buffer was added to the fragmentation mixture and hybridized to the Agilent Mouse GE 4x44K V2 microarray kit (G4846A, Agilent Technologies) for 17 h at 65 °C in a rotating hybridization oven. After hybridization, the microarrays were washed 1 min at RT with GE Wash Buffer 1 and 1 min at 37 °C with GE Wash buffer 2 (Gene Expression Wash Buffer Kit, p/n 5188-5327 Agilent Technologies). Immediately after being washed, the slides were scanned immediately after washing on the Agilent DNA Microarray Scanner (G2505C, Agilent Technologies) using the AgilentHD_GX_1Color Profile (Scan Area: 61 × 21.6 mm; Scan resolution: 5 μm, dye channel set to 100% Green PMT) of the Agilent ScanControl software 8.1.3 (Agilent Technologies). The scanned images were analyzed with Feature Extraction Software 10.7.3.1 (Agilent Technologies) using default parameters (protocol GE1_107_Sep09). Expression data were normalized using quantile normalization. DEGs were computed using the eBay test from the limma R package (v3.48.0). Gene expression data with a false discovery rate (FDR) of ≤0.05 were considered significant. Gene set enrichment analysis (GSEA) was performed using the clusterProfiler R package (v4.0.0) to identify sets of related genes altered in each experimental group.

For RNA-seq analysis, transcript expression quantification was performed using Salmon (v1.9.0) [https://doi.org/10.1038/nmeth.4197] on transcripts annotated in Ensembl 102 (mouse genome reference GRCm38). Gene expression levels were estimated with tximport R package (v1.22.0). We kept all genes with more than 100 reads in at least 2 samples. Principal component analysis was performed on variance stabilized data (vst function from DESeq2 R package v1.34.0 [https://doi.org/10.1186/s13059-014-0550-8]) using prcomp function on the top 5000 most variable genes. DEGs were computed using the DESeq2 R package. DEGs were those genes with adjusted $p \leq 0.05$ and absolute log fold chage ≥1. We performed gene set enrichment analysis using enrichGO from clusterProfiler R package (v4.2.2) and Gene Ontology as gene set source.

### Statistical analysis

To compare the mean difference of a dependent variable between independent groups, two-sample t-tests and one-way analysis of variance (ANOVA), were used for two and more than two groups, respectively. Two-way ANOVAs were performed to evaluate the effects of two categorical predictors on a dependent variable. For all ANOVAs, follow-up Tukey's honest significant difference post hoc tests were conducted for pairwise comparisons. To compare Ca2+ release from FDB fibers using electrical stimulation between groups, we used linear mixed-effect models (LMM) with the fiber size as the dependent variable, mouse as a random effect, measurement and genotype as fixed-effects, and included a random intercept and random slopes for each measurement for each mouse using the afex package in R. For all tests, the significance threshold was set at $p < 0.05$.

### Reporting summary

Further information on research design is available in the Nature Portfolio Reporting Summary linked to this article.

## Data availability

All data generated or analyzed during this study are included in this article and its Supplementary Information files. All requests for raw data and materials should be addressed to the corresponding author. Source data are provided with this paper. Raw RNA-seq FASTQ files and counts data and microarray have been deposited in the Gene Expression Omnibus (GEO) database under accession code GSE221480, and GSE220080, respectively. Source data are provided with this paper.

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

## Acknowledgements

The authors thank Carlo Reggiani and Paolo Bernardi for insightful discussions. We thank Morena Simonato for technical support. This work was supported by Fondazione Telethon (GGP19128 to M.Pe), Association Française contre les Myopathies (22221 to M.Pe and M.B.), PROGRAM RARE DISEASES CNCCS-Scarl-Pomezia (to M.Pe), Kennedy's Disease Association Research Grant (U-GOV PIRA_EPPR19_01 to MPi and AWD00002349 to E.Z.), US National Institutes of Health (R01 NS055746 to A.P.L.), and the Fondazione Umberto Veronesi Fellowship (to E.Z.). S.N. is supported by a Rackham Predoctoral Fellowship and C.R. is supported by a Wellcome Trust Clinical Research Career Development Fellowship (205162/Z/16/Z). G.J.B. was supported by the National Institute of Arthritis and Musculoskeletal and Skin Diseases, US National Institutes of Health (NIH) grant [AR069107]. The Neuromuscular Bank of Tissues and DNA samples, a member of the Telethon Network of Genetic Biobanks (project no. GTB12001) that is funded by Telethon Italy and of the EuroBioBank Network, provided us with biopsy specimens. The HTS and Validation Core Facility at CIBIO (University of Trento, Italy) performed the microarray experiment. EP and GS are members of the European Reference Network for Neuromuscular Diseases—Project ID No 870177. The schemes (Figs. 1 and 8) were created with BioRender.com (CO24QEV61X, JJ24QERHG0, BF24RAQZHX).

## Author contributions

C.Mar., M.Pi. and M.Pe. conceived the study. C.Mar. performed behavioral evaluation, biochemistry and molecular biology experiments, data analysis, and interpretation. G.Z. and M.Pi. contributed to the study design and histological analysis. G.G., C.Mam., S.R., A.R., M.Pa. performed the seahorse analysis and contributed to the calcium experiments. R.A. contributed to the analysis of gene expression. A.Ma. provided the calcium probe. C.Mar., G.Z., M.Pi., A.Me. performed electrophysiological analyses. M.Ch. prepared samples for microarray analysis. N.SR and L.AP performed experiments in knock-in mice. L.N., M.Ca., L.M., and B.B. analyzed muscle force and Ca$^{2+}$ dynamics. L.S. and M.A.D. analyzed mitochondrial respiratory complex activity. E.B., A.P., C.B., P.R., G.S., E.P. provided human muscle biopsies. G.J.B. performed Western blotting to detect SLN expression. P.M. and C.R. performed the bioinformatic analysis. M.Bm. M.S., E.Z. contributed to the study design and data interpretation. C.Mar. and M.Pe. wrote the manuscript. M.Pe. provided the main financial support for this work.

## Competing interests

The authors declare no competing interests.

## Additional information

¹Department of Biomedical Sciences (DBS), University of Padova, 35131 Padova, Italy. ²Veneto Institute of Molecular Medicine (VIMM), Padova 35100, Italy.
³Padova Neuroscience Center (PNC), Padova 35100, Italy. ⁴Dulbecco Telethon Institute (DTI) at the Department of Cellular, Computational and Integrative
Biology (CIBIO), University of Trento, 38123 Trento, Italy. ⁵CIR-Myo, Centro Interdipartimentale di Ricerca di Miologia, University of Padova, 35131 Padova, Italy.
⁶Department of Molecular and Translational Medicine, University of Brescia, 25121 Brescia, Italy. ⁷Department of Pathology, University of Michigan Medical
School, Ann Arbor, MI, USA. ⁸Department of Pharmaceutical and Pharmacological Sciences, University of Padova, 35131 Padova, Italy. ⁹Clinical Genetics Unit,
Department of Women and Children's Health, University of Padova, and Fondazione Istituto di Ricerca Pediatrica Città della Speranza, Padova, Italy.
¹⁰Department of Neuroscience (DNS), University of Padova, 35128 Padova, Italy. ¹¹Orthopedics and Orthopedic Oncology, Department of Surgery, Oncology,
and Gastroenterology DiSCOG, University-Hospital of Padova, 35128 Padova, Italy. ¹²Musculoskeletal Pathology and Oncology Laboratory, Department of
Surgery, Oncology and Gastroenterology (DiSCOG), University of Padova, 35128 Padova, Italy. ¹³Department of Biology, University of Padova, Padova 35100,
Italy. ¹⁴Department of Cell Biology and Molecular Medicine, Rutgers, New Jersey Medical School, Newark, NJ 07103, USA. ¹⁵Department of Cellular,
Computational and Integrative Biology (CIBIO), University of Trento, 38123 Trento, Italy. ¹⁶These authors contributed equally: Caterina Marchioretti, Giulia
Zanetti, Marco Pirazzini. ✉e-mail: maria.pennuto@unipd.it

