## [Peer Review File · Nature Communications]

Defective excitation-contraction coupling and mitochondrial respiration precede mitochondrial Ca²⁺ accumulation in spinobulbar muscular atrophy skeletal muscleReviewers' comments:

Reviewer #1 (Remarks to the Author):

Marchioretta, Zanetti et al. aim to explore early disease pathways driving skeletal muscle decline in Spinal and Bulbar Muscular Atrophy (SBMA), caused by a CAG repeat expansion of the Androgen Receptor (AR) gene. This manuscript uses microarray analysis of mouse models and SBMA patient biopsies to definitively implicate Excitation-Contraction Coupling (ECC) machinery dysregulation in skeletal muscle of SBMA mice. While this ECC finding is interesting and well characterized by the authors, the manuscript suffers from excessive phenotypic directions, which at times raises confusion. But more importantly in terms of the impact of the work, this paper does not achieve a substantive mechanistic understanding of the disease process being studied.

Issues that the authors should address to improve the manuscript:

- 1) It is unclear how to compare the use of different SBMA models in this work. How does the 12 week time point of AR113Q used for microarray in Fig. 2 relate to the 4 and 8 week time points of AR100Q mice in Fig. 1 and the rest of the manuscript? In lines 146-147, the authors state that this 12 week time-point of AR113Q “precedes motor dysfunction” but in the referenced publication (7), it is stated that motor deficits “appeared in adults as early as 8 weeks of age” in this model. Please provide clarification on the chosen timepoints.
- 2) RNA-seq data from 14 week old AR113Q quadriceps muscle has already been published by the authors, severely limiting the novelty and relevance of this manuscript's 12-week microarray dataset. The authors should perform transcriptome analysis of the AR100Q model at 4 and 8 weeks.
- 3) Figure 5: The authors draw parallels to PV knockout mice (citation 34), but those mice show robust hypertrophy, which brings into question the authors' interpretation of the alteration of calcium dynamics as “impaired”. Instead of the excessive phenotyping data of all different facets of muscle biology presented in this manuscript, a more targeted and experimental/mechanistic approach towards a single direction, be it calcium dynamics, mitochondrial respiration, muscle architecture, etc., where described players are rescued or silenced is necessary, as the manuscript data presented is mostly descriptive and correlative.
- 4) The rescue work is not compelling. The authors need to show that therapeutic targeting specifically of the described ECC dysfunction (by silencing SLN, restoring PV, or similar) rescues neuromuscular disease. Without such data, the results remain rather superficial.
- 5) Fig 3D: please show full blots of all replicates.
- 6) Fig. 4A: this staining does not look specific. Why is there so little MyHC Type IIB staining in 4 week quadriceps? Why does 8 week Ryr1 staining appear brightest around the periphery of fibers, unlike the other 3 images? Is that typical? Myofibers appear somewhat smaller in 4 week AR100Q muscle. Please include myofiber area measurements in this timepoint or reference in the text if this has been measured in a previous publication using this model.
- 7) Fig. 4B-C: basal respiration in WT myofibers appears higher than AR100Q fibers by Seahorse but the activity of respiratory chain complexes is higher in AR100Q mitochondria, which seems contradictory. Could the authors please explain this discrepancy? It is necessary to assay complex activity in 4 week muscle as well in order to obtain a complete picture of what is going on here.
- 8) The text is too long and at times unfocused and diffuse. The Discussion section should be significantly parsed down, for example. Please also perform proofreading of this section, as there were a number of errors.

Reviewer #2 (Remarks to the Author):

The manuscript entitled “Defective excitation-contraction coupling and mitochondrial respiration precede mitochondrial Ca²⁺ accumulation in spinobulbar muscular atrophy skeletal muscle” by Marchionetti and colleagues aims to identify early pathological processes and how they unfold during the progression of the disease of spinobulbar muscular atrophy (SBMA). The authors use multiple experimental approaches to address important questions in the disease process. The study is elaborate using both mouse models of disease as well as myofibers from human patients. The manuscript is rather convoluted and not easy to follow (neither the rationale, nor the logic of experimental sequence). Lack of attention in several aspects makes cumbersome the comprehension of the main points of the study. Attention is strongly advised in the statistical analysis employed, as well as appropriate explanation of the experimental design. In addition, clarification in the number of observations raises some concerns regarding the depth of analysis, rigor and importance of the findings reported. On the assumption that several major concerns will be addressed properly, this study provides important findings towards the deregulation of ECC gene expression as well as the decreased mitochondrial respiration as being two pathological processes that occur early in SBMA muscle. These findings should be beneficial to our broader understanding of the mechanisms involved in SBMA. The Discussion will be benefit a lot if it is more focused and amplified on the biological relevance and clinical significance of the results from their study.

The following major and some minor concerns need to be addressed:

1. The CMAP results shown in Fig.1b are problematic and require reevaluation and perhaps analysis anew. For instance, although the authors do not explain how they measured the amplitude of the CMAP (i.e. baseline-to-peak or peak-to-peak), irrespective of the measurement method, the voltage scale bar in Fig.1b is 10mV, indicating that the max amplitude (peak-to-peak is approximately 50-60mV), yet, the amplitudes of the CMAPs that are shown in the graphs are of the magnitude of approximately 800mV in the TA muscle and approximately 1000mV for the gastrocnemius. How is this possible? In addition, the authors use a t-test for the middle and right graph in Fig.1b which is not appropriate, since the proper comparison should employ oneWay ANOVA test (three different ages and two genotypes). To this end, in the gastrocnemius graph CMAP amplitude, wild type mice across the three ages appear to have different CMAP amplitude (i.e. 1000mV at 4 weeks of age, ~700mV at 8 weeks of age and ~600mV at 12 weeks of age). Is there a statistical difference between the three WT groups? For this reason, oneWay ANOVA analysis should be used. If there is a statistical difference in CMAP amplitude between the three WT ages, what is the significance of this difference? Does this mean that the CMAP amplitude sensitivity test is approximately 20-30%? That points goes towards the underlying significance at 4 weeks of age in the gastrocnemius CMAP graph, which might not be statistically sound. Additionally, it is unclear what each point in the graph represents. Does each point represents a single muscle from a single mouse or the average of both muscles from a single mouse? If so, was there a test conducted to compare the CMAP from both muscles from a single mouse? Finally, the authors state that the CMAP amplitude is supramaximal but the authors state that this was assumed on the stimulation intensity, rather than stating that supramaximal stimulation was ensured that the CMAP amplitude did not increase any further irrespective of the stimulation intensity used, which is a more accurate method to ensure that all motor neurons were recruited to result in the maximum possible CMAP amplitude. The authors are strongly advised to explain and re-analyze the results from these experiments.

2. How the authors can exclude the possibility that the reduction in the CMAP amplitude from the gastrocnemius at 4 weeks of age (and the subsequent ages – and the same applies for the TA CMAP) may be due to dysfunction of gastrocnemius motor neurons? In other words, gastrocnemius motor neurons cannot elicit an action potential and therefore not being able to activate their muscle fibers? Is the H-reflex observed in this type of experiments? The presence of the H-reflex would provide more confidence that the function of motor neurons is not compromised.

3. It is unclear why the authors opt to study different types of muscles (TA, EDL, gastrocnemius, soleus, quadriceps, FDB etc), especially when the authors state that “SBMA is a progressive late-

onset disease characterized by the selective degeneration of lower motor neurons (MNs)..." (lines 60-61 in the Intro). The authors should clarify which muscles are considered to be selectively vulnerable in SBMA and explain their rationale for each muscle studied for each experiment. Furthermore, the authors should provide adequate explanation of the different muscles studied in their study and how relevant are these muscles to disease phenotype.

4. It is unclear for this reviewer how lower relaxation time caused a shift to the left of the normalized force-frequency curve (as shown in Fig.1f). The authors are advised to clarify their interpretation of this result. Furthermore, the authors do not explain how they normalized the force under different frequency of stimulation. The authors are required to perform statistical analysis (using ANOVA) for the normalized force in fig.1f for the EDL experiments. What is the biological significance of the increase in force in the AR100Q mice (compared to WT controls) at frequencies 20-90Hz (at 4 weeks of age) and 30-60Hz (at 8 weeks of age)? Finally, it is unclear how the authors obtain a force with no stimulation frequency (i.e. the first data point in all tables in fig.1f), or is it that the authors used a very low frequency of stimulation that appears as if the stimulation frequency was zero?

5. In order to, "elucidate the molecular processes underlying structural and functional changes responsible for muscle force reduction", the authors adopted an unbiased approach and performed a transcriptomic analysis. To do so, the authors performed the analysis in 12-week-old mice, (since, "knock-in mice corresponds to the presymptomatic stage"). However, in lines 161-164, the authors state: "By real-time PCR analysis, we found that the transcript levels of key ECC genes, including *Cacna1s*, which encodes a subunit of DHPR, *Ryr1*, *Atp2a1*, *Casq1* and *Pv*, were significantly decreased, while those of *Casq2* and *Sln* were up-regulated in the quadriceps, EDL, and flexor digitorum brevis (FDB) muscles of 8-week-old transgenic SBMA mice". Why the two different ages (i.e. 12 weeks and 8 weeks)? This is confusing and the authors should clarify this apparent mismatch in age.

6. Statistical analysis for results presented in Fig.3a used t-test. However, the authors should perform oneWay ANOVA statistical comparison and report significance across the two different ages for the genes studied.

7. Perhaps I am mistaken, but the results for the *Atp2a1* shown in Fig.3b are not the same (or as expected) when compared with the results in Fig.3a, where *Atp2a1* is significantly reduced both at 4 weeks and 8 weeks of age. Could the authors explain this discrepancy?

8. The immunohistochemistry in Fig.4a (for the 8 weeks old WT mice) is not convincing and it appears problematic for the RYR1 immunoreactive signal. The fluorescence signal appears to be diffuse and very weak within the myofibers and nearly equal to that of the background signal which is expected to be nearly black. The authors are encouraged to present better images for this important control. In addition, it appears that all fibers are positive for MyHC type IIb.

9. Regarding the experiments on the oxygen consumption rate (OCR), the OCR appears to be lower in the SBMA muscles both in 4 and 8 week old muscles (as shown in both graphs prior to the application by oligomycin). Is this difference statistically significant? The authors did not report any statistical comparison for the Seahorse analysis. Irrespective of the statistical difference, could the authors provide any explanation of the difference in OCR between WT and SBMA muscles?

10. The authors state (lines 199-200) "Interestingly, the transcript levels of *Nrf1* were downregulated, whereas those of *Nrf2*, *Sdha*, and *Tfam* were upregulated in the skeletal muscle of 8-week-old AR100Q mice compared to control mice (Fig. 4d)." Could the authors expand on the significance of these observations? I fail to understand the importance of the apparent differential regulation of *Nrf1* (downregulation) and *Nrf2* (upregulation).

11. Based on the results shown in Fig.3a regarding *Ryr1* mRNA levels, in which AR100Q muscles reveal significant lower levels compared to WT muscles, is not surprising that the authors did not detect any lower immunoreactive signals of *Ryr1* (in conjunction with the immunoreactivity against

translocase of outer mitochondrial membrane 20 (TOM20), as shown in Fig.6b. Wouldn't the authors expect a significant lower signal of Ryr1 in AR100Q muscles compared to WT muscles?

12. The authors should perform statistical testing for the results shown in Fig.7a and reports the results in the graph.

Minor concerns

1. In Fig.1d, the authors state that "n=4-6mice/genotype", however in the TETANUS graph, there are only 3 data points in the WT bar. The authors should correct the "n" description.

2. Details of anesthesia are missing for the CMAP experiments.

3. It will be very helpful to illustrate with traces the force from a twitch and tetanus in the EDL experiments shown in Fig.1c.

4. The authors should use a different loading control (other than calnexin) in the Westerns presented in Fig.1d, since it is clear that calnexin is not uniform in neither mice or human samples, raising some concerns about the interpretation of the results.

Reviewer #3 (Remarks to the Author):

NCOMMS-22-00863

Spinal and bulbar muscular atrophy (SBMA) is a progressive late-onset motor neuron disease caused by abnormal CAG repeat expansions in the androgen receptor (AR) gene. Recent evidences suggest that not only motor neuron but also skeletal muscle are primary contributors to disease pathogenesis in SBMA. Pennuto and her colleague here demonstrated early and late events in SBMA muscles. The main findings of this study are dysregulation of the excitation-contraction coupling (ECC) machinery and mitochondrial dysfunction in the skeletal muscle of SBMA mouse models at an early stage. This finding is important to elucidate the muscle pathophysiology in SBMA. The paper is well written and compelling. The reviewer's concerns are listed below.

Major points:

1. Early defects in mitochondrial respiration in SBMA muscle (Fig. 4), which are reversed by castration (Fig. 7a), are interesting findings of this study. Mitochondrial Ca²⁺ accumulation in the skeletal muscle of 8-week-old AR100Q mice (Fig. 5), which might lead to mitochondrial dysfunction and mitophagy, is also interesting. Confirmation of altered expression of oxidative stress response genes and mitochondrial genes (Figs. 4d and 4e) in patient biopsy samples should strengthen the evidence of these important findings.

2. Muscle contraction/relaxation and force, expression of key excitation-contraction coupling (ECC) genes, and myofiber respiration were significantly affected as early as 4 weeks of age in AR100Q transgenic mice (Figs. 1e, 1f, 3a, 4b), and notably, these defects were consistently observed between 4-week and 8-week-old muscles. Please provide insights into this potential developmental and androgen-independent pathogenesis.

3. The authors demonstrated altered gene expression in sarcomere organization and muscle contraction in pre-symptomatic AR113Q mice (Fig. 2). The authors may want to comment on whether these alterations related to deregulation of ECC machinery or mitochondrial dysfunction.

4. Furthermore, some of ECC genes are expressed in a fiber-type specific manner. For instance, Atp2a1 and Casq1 are dominantly expressed in fast-twitch muscles, and Atp2a2 and Casq2 in slow-twitch muscles. Ref 24 showed fast- to slow-twitch muscle fiber-type switching was observed as early

as 40-day-old in AR113Q mice. These alterations of ECC genes in the skeletal muscle of SBMA mouse models and patients might be the result of muscle fiber-type switching rather than mutant AR-induced transcriptional change. Indeed, induction of mutant AR by Dox had relatively little effects on expression of Atp2a1 and Casq1 genes (Fig. 3b). This needs some discussion.

Minor points:

1. GO analysis 'cell component' (P5, line 149), 'Biological processes' (line 153), and 'Molecular functions' (line 155) should be 'Cellular component' and 'Biological process', and 'Molecular function', respectively.

2. On P7, line 223, correct "whichl".

3. In the upper part of Fig. 8, schematics of SR and mitochondria in physiological condition and SBMA are described, but explanation is lacking. It may be preferable that the authors describe explanation of the schematics in Discussion or the figure legend.

Reviewer #4 (Remarks to the Author):

This manuscript is an attempt to describe the time course of events leading to the development of spinobulbar muscular atrophy. To that aim, they used a knock-in mice model showing that altered androgen receptor (AR) alters the generation of muscle force prior to denervation. The pattern of gene expression in the mice model mimics that of tissue from patients with the disease and it is interesting that several genes of the excitation-contraction process are downregulated. These findings correlate with altered morphology of the muscle triad. Another finding is the early alteration of mitochondrial respiration and mitochondria calcium content. The early pathological process can be prevented by AR silencing or surgical castration.

The ensemble of results suggests that altered androgen receptors in skeletal muscle participate in a series of transcriptional events that will lead to the development of this disease of which many things remain unknown. This work has an original approach that may facilitate future studies in this interesting area.

Point-by-point response to the reviewers' comments:

Reviewer #1 (Remarks to the Author):

Marchioretta, Zanetti et al. aim to explore early disease pathways driving skeletal muscle decline in Spinal and Bulbar Muscular Atrophy (SBMA), caused by a CAG repeat expansion of the Androgen Receptor (AR) gene. This manuscript uses microarray analysis of mouse models and SBMA patient biopsies to definitively implicate Excitation-Contraction Coupling (ECC) machinery dysregulation in skeletal muscle of SBMA mice. While this ECC finding is interesting and well characterized by the authors, the manuscript suffers from excessive phenotypic directions, which at times raises confusion. But more importantly in terms of the impact of the work, this paper does not achieve a substantive mechanistic understanding of the disease process being studied. Issues that the authors should address to improve the manuscript:

1) It is unclear how to compare the use of different SBMA models in this work. How does the 12 week time point of AR113Q used for microarray in Fig. 2 relate to the 4 and 8 week time points of AR100Q mice in Fig. 1 and the rest of the manuscript? In lines 146-147, the authors state that this 12 week time-point of AR113Q “precedes motor dysfunction” but in the referenced publication (7), it is stated that motor deficits “appeared in adults as early as 8 weeks of age” in this model. Please provide clarification on the chosen timepoints.

We thank the reviewer for raising this point. We added a Table (Table 1, shown in the main text of the revised manuscript) to compare the most relevant features of the different mouse models used in this study. In the manuscript reporting the original characterization of the phenotype of the SBMA knock-in mice, motor dysfunction appeared at 8 weeks of age, as pointed out by this reviewer¹. However, over the years we observed a delay in the onset of motor dysfunction in the knock-in mice. We thus phenotyped the knock-in line again, and we established that 12 weeks of age precedes motor dysfunction, which is overt around 20 weeks of age^{2,3,4}. We modified the text accordingly to clarify this point, and we added the proper references.

2) RNA-seq data from 14 week old AR113Q quadriceps muscle has already been published by the authors, severely limiting the novelty and relevance of this manuscript's 12-week microarray dataset. The authors should perform transcriptome analysis of the AR100Q model at 4 and 8 weeks.

To address this point, we performed RNA seq analysis in the quadriceps muscle of WT and AR100Q mice at 4 and 8 weeks of age. By comparing the transcriptomics of AR100Q mice and their wild type siblings, we found that the expression of several genes was dysregulated starting at 4 weeks of age, further supporting the idea that transcription dysregulation is a key pathological process that precedes disease manifestations in mice. Our gene ontology analysis revealed that the main altered pathways were muscle contraction and sarcomere structure. In particular, the expression of several ECC genes was altered by 4 weeks of age. Gene expression dysregulation is exacerbated at 8 weeks of age, consistent with disease progression. This piece of data is important, as it shows that ECC gene expression is

compromised very early, long before motor dysfunction. This analysis validates our main conclusion that early dysregulation of ECC gene expression is a key pathological event in SBMA muscle. These new results are shown in Fig. 2 of the revised manuscript.

3) Figure 5: The authors draw parallels to PV knockout mice (citation 34), but those mice show robust hypertrophy, which brings into question the authors' interpretation of the alteration of calcium dynamics as "impaired". Instead of the excessive phenotyping data of all different facets of muscle biology presented in this manuscript, a more targeted and experimental/mechanistic approach towards a single direction, be it calcium dynamics, mitochondrial respiration, muscle architecture, etc., where described players are rescued or silenced is necessary, as the manuscript data presented is mostly descriptive and correlative.

We agree with this reviewer that the phenotype of *Pv* knockout mice is different from the phenotype of SBMA mice, likely because SBMA muscles are characterized by the loss of several ECC genes. Here we show two previously unexplored pathological processes occurring early (at presymptomatic stage) in SBMA mice and that are rescued by decreasing the toxicity of the disease protein with two independent approaches, namely androgen deprivation and AR silencing. The ECC process and mitochondrial function are intimately linked to each other, structurally and functionally, so that damage in one process is likely to affect the other. One aspect that is particularly relevant is that *Pv* knock-out mice show accumulation of Ca^{2+} into mitochondria, supporting how the ECC process is linked to the role that mitochondria play in the process of adaptation to stress. It is possible that etiologically SBMA mitochondria are unable to compensate for long time the defects of the ECC process, and this may occur long (months in mice and years in patients) before the onset of motor dysfunction in patients. We modified the text accordingly (the paragraph related to Fig. 5 in the Results section, and the paragraph of the Discussion section, page 10).

4) The rescue work is not compelling. The authors need to show that therapeutic targeting specifically of the described ECC dysfunction (by silencing SLN, restoring PV, or similar) rescues neuromuscular disease. Without such data, the results remain rather superficial.

We performed transcript level analysis of the ECC genes before the rising of androgen levels in mice, which occurs at 4 weeks of age. We found that the expression of ECC genes is normal in the muscle of 11- and 20-day-old SBMA mice (Fig. 3a). Their dysregulation occurs at 4 weeks of age, which is consistent with the rise in androgen levels occurring in mice ⁵. We thus reasoned that modification of a single gene may not result in modification of SBMA phenotype. Rather, we opted for assessing the effect of the current treatment of SBMA patients on the early pathological processes identified here. SBMA is an androgen-dependent disease, and the only authorized (in Japan) medication consists in androgen deprivation, which in mice can be recapitulated with surgical castration ⁶. Another promising approach under development consists in the silencing of AR ². We show that both approaches rescue ECC gene expression changes and mitochondrial respiration. Based on the nature of SBMA, we believe that this finding is extremely important, as it shows that these early pathological processes are reversible. We were not

clear enough in the original version of the manuscript. Thus, we added a scheme (Fig. 7a) to emphasize that castration is performed when the ECC gene expression changes are already present. Moreover, we added a paragraph to the Discussion section to highlight the relevance of these findings, mainly two points:

- i) **ECC gene expression, dynamics of muscle contraction, fatigue and myofiber respiration were altered at presymptomatic stage (by 4 weeks of age); notably, by performing gene expression analysis, we found that ECC gene expression was normal by 20 days of age (Fig. 3a of the revised manuscript), indicating that their dysregulation is concomitant with androgen rise in the serum, which starts at 4 weeks of age.**
- ii) **These early pathological processes are reversible, which reinforces the idea that androgen deprivation is a clinical intervention (for the moment approved only in Japan) that should be extended to other Countries and that may be associated with additional treatment designed to increase muscle mass and force (anabolic signaling) or to decrease muscle weakness and atrophy (for instance, by targeting the early pathways dysregulated in muscle, as shown in our manuscript).**

5) Fig 3D: please show full blots of all replicates.

We show now the entire dataset of replicates as requested. These data are shown in Figure 3d.

6) Fig. 4A: this staining does not look specific. Why is there so little MyHC Type IIB staining in 4 week quadriceps? Why does 8 week Ryr1 staining appear brightest around the periphery of fibers, unlike the other 3 images? Is that typical? Myofibers appear somewhat smaller in 4 week AR100Q muscle. Please include myofiber area measurements in this timepoint or reference in the text if this has been measured in a previous publication using this model.

We reanalyzed the data and performed new analysis in 4-week-old and 8-week-old WT and SBMA mice (3 additional mice/genotype and age). We obtained results that now clearly show that RYR1 staining is lost in the center of type IIb fibers of SBMA mice at 8 weeks of age, and not at 4 weeks of age. We have already measured the cross-sectional area (CSA) of glycolytic and oxidative myofibers of the quadriceps muscle of 4- and 8-week-old WT and AR100Q mice, and these data have been previously published ⁷. As suggested by this reviewer, we added the reference to the manuscript.

7) Fig. 4B-C: basal respiration in WT myofibers appears higher than AR100Q fibers by Seahorse but the activity of respiratory chain complexes is higher in AR100Q mitochondria, which seems contradictory. Could the authors please explain this discrepancy? It is necessary to assay complex activity in 4 week muscle as well in order to obtain a complete picture of what is going on here.

To address this point, we measured mitochondrial complex activity in 4- and 8-week-old AR100Q and wild type mice. We normalized the data of complex activity to citrate synthase activity, and we performed statistical analysis (one-way ANOVA followed by Tukey post-hoc test). In the original manuscript, we performed t test statistical analysis, but as indicated by the reviewers, we have

now used ANOVA for multiple comparisons and performed new statistical analysis. We found that complex activity was normal at both 4 and 8 weeks of age. These data are presented in Fig 4c of the revised manuscript. In the Discussion, we added a paragraph to comment that defects in mitochondrial respiration are not due to defects in complex activity (paragraph before the last).

8) The text is too long and at times unfocused and diffuse. The Discussion section should be significantly parsed down, for example. Please also perform proofreading of this section, as there were a number of errors.

We have carefully followed the reviewer suggestions. We shortened the text and in particular the Discussion section to have it more focused on the major findings reported in this manuscript, namely the early deregulation of the ECC process, dynamics of muscle contraction, fatigue and myofiber respiration. Moreover, the final text is being edited by American J. Experts (AJE) Digital Editing (<https://www.aje.com/services/digital/>).

Reviewer #2 (Remarks to the Author):

The manuscript entitled “Defective excitation-contraction coupling and mitochondrial respiration precede mitochondrial Ca²⁺ accumulation in spinobulbar muscular atrophy skeletal muscle” by Marchionetti and colleagues aims to identify early pathological processes and how they unfold during the progression of the disease of spinobulbar muscular atrophy (SBMA). The authors use multiple experimental approaches to address important questions in the disease process. The study is elaborate using both mouse models of disease as well as myofibers from human patients. The manuscript is rather convoluted and not easy to follow (neither the rationale, nor the logic of experimental sequence). Lack of attention in several aspects makes cumbersome the comprehension of the main points of the study. Attention is strongly advised in the statistical analysis employed, as well as appropriate explanation of the experimental design. In addition, clarification in the number of observations raises some concerns regarding the depth of analysis, rigor and importance of the findings reported. On the assumption that several major concerns will be addressed properly, this study provides important findings towards the deregulation of ECC gene expression as well as the decreased mitochondrial respiration as being two pathological processes that occur early in SBMA muscle. These findings should be beneficial to our broader understanding of the mechanisms involved in SBMA. The Discussion will benefit a lot if it is more focused and amplified on the biological relevance and clinical significance of the results from their study. The following major and some minor concerns need to be addressed:

1. The CMAP results shown in Fig.1b are problematic and require reevaluation and perhaps analysis anew. For instance, although the authors do not explain how they measured the amplitude of the CMAP (i.e. baseline-to-peak or peak-to-peak), irrespective of the measurement method, the voltage scale bar in Fig.1b is 10mV, indicating that the max amplitude (peak-to-peak is approximately 50-60mV), yet, the amplitudes of the CMAPs that are shown in the graphs are of the magnitude of approximately 800mV in the TA muscle and approximately 1000mV for the gastrocnemius. How is this possible?

We reanalyzed all the original CMAP data, as suggested by this reviewer. Moreover, we extended the analysis to other mice to increase the number of observations, and we confirmed our original findings. We clarified that we measured the peak-to-peak amplitude, and we added this information to the Methods section. Furthermore, in the original manuscript we did not calibrate the data, and we apologize for this oversight. We have now normalized the CMAP amplitude using 1 Volt that correspond to 12 mV amplitude in our setup. This calibration resulted in a peak-to-peak value around 50-60 mV, as indicated by this reviewer.

In addition, the authors use a t-test for the middle and right graph in Fig.1b which is not appropriate, since the proper comparison should employ oneWay ANOVA test (three different ages and two genotypes). To this end, in the gastrocnemius graph CMAP amplitude, wild type mice across the three ages appear to have different CMAP amplitude (i.e. 1000mV at 4 weeks of age, ~700mV at 8 weeks of age and ~600mV at 12 weeks of age). Is there a statistical difference between the three WT groups? For this reason, oneWay ANOVA analysis should be used. If there is a statistical difference in CMAP amplitude between the three WT ages, what is the

significance of this difference? Does this mean that the CMAP amplitude sensitivity test is approximately 20-30%? That points goes towards the underlying significance at 4 weeks of age in the gastrocnemius CMAP graph, which might not be statistically sound.

We thank this reviewer for making this point. We have performed new statistical analysis. To compare the two genotypes with each other and by age, we performed two-way ANOVA followed by Tukey post-hoc test. We did not find any statistical difference in the WT mice across age, while the CMAP of TA and gastrocnemius muscles of AR100Q at 8 and 12 weeks of age was different from that of the other groups. The results of the analysis are presented in Fig. 1b and Supplementary Fig. 1a of the revised manuscript.

To only assess whether there is a change in the CMAP of WT mice by age, we applied the one-way ANOVA followed by Tukey post-hoc test, as shown in the panels below. We confirm that CMAP was not significantly different in the WT mice across age.

Additionally, it is unclear what each point in the graph represents. Does each point represent a single muscle from a single mouse or the average of both muscles from a single mouse? If so, was there a test conducted to compare the CMAP from both muscles from a single mouse?

In the original manuscript, each dot was a single measure, which was repeated three times for each mouse (9 dots from 3 mice). In the revised manuscript, after increasing the animal numerosity, each point represents the average of at least 3 CMAP amplitudes from a 1-minute registration of each muscle analyzed. We recorded CMAP on gastrocnemius and tibialis anterior muscles of the same animal. This information was added to the Methods section.

Finally, the authors state that the CMAP amplitude is supramaximal but the authors state that this was assumed on the stimulation intensity, rather than stating that supramaximal stimulation was ensured that the CMAP amplitude did not increase any further irrespective of the stimulation intensity used, which is a more accurate method to ensure that all motor neurons were recruited to result in the maximum possible CMAP amplitude. The authors are strongly advised to explain and re-analyze the results from these experiments.

In the original manuscript this experiment was not exhaustively explained. We performed two types of analysis: a supramaximal stimulation and a constant stimulation at 15 V. In the revised manuscript, according to the reviewer's comment, we reported results obtained from supramaximal stimulation. We apologize for omitting this information in the original manuscript.

2. How the authors can exclude the possibility that the reduction in the CMAP amplitude from the gastrocnemius at 4 weeks of age (and the subsequent ages – and the same applies for the TA CMAP) may be due to dysfunction of gastrocnemius motor neurons? In other words, gastrocnemius motor neurons cannot elicit an action potential and therefore not being able to activate their muscle fibers? Is the H-reflex observed in this type of experiments? The presence of the H-reflex would provide more confidence that the function of motor neurons is not compromised.

Following the reviewer suggestion, we performed H-reflex analysis in AR100Q and WT mice at 4 weeks, 8 weeks, and 12 weeks of age. FDB muscle EMG recordings were acquired by stimulating the sciatic nerve as already described in Materials and Methods of submitted manuscript. Stimulation intensity was set at around 1/3 of the supramaximal stimulation⁸. We used FDB where we could obtain a clear H-reflex signal and a neat resolution of the H-wave. We detected the H-reflex in all analyzed muscles, and we did not observe any difference in the H-reflex latency of AR100Q mice compared to WT siblings, confirming that the motor neuron functionality is preserved. These results are presented in Supplementary Fig. 1b of the revised manuscript.

3. It is unclear why the authors opt to study different types of muscles (TA, EDL, gastrocnemius, soleus, quadriceps, FDB etc), especially when the authors state that “SBMA is a progressive late-onset disease characterized by the selective degeneration of lower motor neurons (MNs)...” (lines 60-61 in the Intro). The authors should clarify which muscles are considered to be selectively vulnerable in SBMA and explain their rationale for each muscle studied for each experiment. Furthermore, the authors should provide adequate explanation of the different muscles studied in their study and how relevant are these muscles to disease phenotype.

We thank the reviewer for making this point. We specified that our experiments were performed in different types of skeletal muscle, including muscles that are mainly glycolytic (quadriceps, gastrocnemius, EDL, TA, and FDB) and a muscle that is mainly oxidative (soleus), based on our previous finding that glycolytic muscles are more severely affected in SBMA compared to oxidative muscles^{7,9,10}. Here we addressed whether different types of glycolytic muscles show different degrees of severity with respect to force, dynamics of contraction and dysregulation of gene expression. We evaluated at least two types of glycolytic muscles where this was possible, yet always a glycolytic muscle versus an oxidative muscle. For technical reasons, some experiments were performed in specific types of skeletal muscle (e.g., ex-vivo electrical stimulation in ADL & soleus, calcium imaging in FDB-isolated fibers). For this reason, we first confirmed that each single type of muscle used in this study show ECC deregulation, which is an essential observation to justify the use of that type of muscle for further analysis. We added a section to the Discussion to better explain the rationale for the use of different types of muscles (second paragraph).

4. It is unclear for this reviewer how lower relaxation time caused a shift to the left of the normalized force-frequency curve (as shown in Fig.1f). The authors are advised to clarify their interpretation of this result.

We apologize for the lack of clarity in the original manuscript. The slowing of contractile speed associated with muscle fatigue causes a shift of the force–frequency curve to the left, which allows the tetanic fusion of force to occur at lower firing frequencies. Thus, the increased relaxation time indicates that SBMA fibers rich normalized maximal force in response to lower frequency stimulation. For the sake of clarity, we added an explanation of interpretation of data.

Furthermore, the authors do not explain how they normalized the force under different frequency of stimulation.

We apologize for the lack of clarity here. Force was normalized to the maximum force to align the curves and determine as to whether there is a shift in the curves. We added this information to the Methods section.

The authors are required to perform statistical analysis (using ANOVA) for the normalized force in fig.1f for the EDL experiments.

We performed two-way ANOVA followed by Tukey post-hoc test by comparing genotypes (WT vs SBMA) and frequency stimulation. Statistical significance is now shown in the figure panel.

What is the biological significance of the increase in force in the AR100Q mice (compared to WT controls) at frequencies 20-90Hz (at 4 weeks of age) and 30-60Hz (at 8 weeks of age)?

In panel 1f, we measured the maximal force that can be obtained from the analyzed muscle. What we are observing here is that we reach maximum force at lower frequency stimulation, strongly indicating that, although muscle force in SBMA is lower compared to WT mice, its maximum is reached with lower frequency stimulation. The biological significance of this finding is that SBMA glycolytic muscles rich fatigue faster compared to wild type mice. This is important, as patients complain that fatigue is one of the earliest symptoms throughout life, often used to set disease onset. We added this information in the main text of the revised manuscript (second paragraph of the Discussion section).

Finally, it is unclear how the authors obtain a force with no stimulation frequency (i.e. the first data point in all tables in fig.1f), or is it that the authors used a very low frequency of stimulation that appears as if the stimulation frequency was zero?

We apologize for omitting this information in the original manuscript. The first stimulation is at 1 Hz, but for the resolution of the graph and the figure, it is difficult to visualize it. We have now started the X axis from 1, and for the sake of clarity we added 1 to the X axis of each panel. We thank this reviewer for pointing this out.

5. In order to, “elucidate the molecular processes underlying structural and functional changes responsible for muscle force reduction”, the authors adopted an unbiased approach and performed a transcriptomic analysis. To do so, the authors performed the analysis in 12-week-old mice, (since, “knock-in mice corresponds to the presymptomatic stage”). However, in lines 161-164, the authors state: “By real-time PCR analysis, we found that the transcript levels of key ECC genes, including *Cacna1s*, which encodes a subunit of DHPR, *Ryr1*, *Atp2a1*, *Casq1* and *Pv*, were

significantly decreased, while those of Casq2 and Sln were up-regulated in the quadriceps, EDL, and flexor digitorum brevis (FDB) muscles of 8-week-old transgenic SBMA mice". Why the two different ages (i.e. 12 weeks and 8 weeks)? This is confusing and the authors should clarify this apparent mismatch in age.

To better clarify the rationale for the ages of each SBMA mouse model used in this study, we added a Table (Table 1, please, see also response to Reviewer 1.1). We also provide a schematic of the ages analyzed here, including a time point before puberty (20 days of age).

To further corroborate our gene expression analysis, we performed new transcriptomic (RNAseq) analysis (suggested by reviewer 1) in AR100Q mice and WT siblings at 4 and 8 weeks of age (Fig. 2 of the revised manuscript. Please, see also response to reviewer 1 point 2). By RNAseq we found early changes in ECC gene expression that augmented by 8 weeks of age. The major affected pathways were related to muscle contractile proteins and sarcomere function. These results show that several genes are dysregulated very early, before motor dysfunction, suggesting that transcription dysregulation is a primary component of muscle pathology in SBMA.

6. Statistical analysis for results presented in Fig.3a used t-test. However, the authors should perform oneWay ANOVA statistical comparison and report significance across the two different ages for the genes studied.

Due to the different expression levels of the ECC genes in physiological conditions (WT mice) at different ages (4 versus 8 weeks of age), we normalized the data to the results obtained in SBMA mice versus those obtained in WT mice. Because our goal here is to determine whether the levels of expression of specific ECC genes between SBMA and WT mice change at a specific age, we separated the results obtained at 4 weeks of age and 8 weeks of age, and we performed t test analysis. We also added new data obtained at 20 days of age, which precedes androgen rise in the serum. Notably, ECC gene expression changes were absent at p20, and they were detectable starting at p30. This strongly suggests that these ECC gene expression changes are triggered by androgens in myofibers expressing polyQ-expanded AR. Consistent with this idea, surgical castration and AR silencing performed after testosterone levels pick to their plateau in the serum restore these alterations.

7. Perhaps I am mistaken, but the results for the Atp2a1 shown in Fig.3b are not the same (or as expected) when compared with the results in Fig.3a, where Atp2a1 is significantly reduced both at 4 weeks and 8 weeks of age. Could the authors explain this discrepancy?

We thank this reviewer for raising this point. We have specified that in iAR100Q mice three ECC genes were found to be dysregulated, namely CASQ1, Pv, and Sln. For the sake of clarity, we added all the analyzed ECC genes to the main (Fig. 3). There are at least two reasons as to why only three ECC genes are dysregulated in iAR100Q mice: i) because the expression of specific isoforms of ECC genes is fiber type-specific, it is possible that at least in part these gene expression changes are the result of glycolytic-to-oxidative fiber-type switch occurring in AR100Q mice, but not in iAR100Q mice; ii) Possibly, expression of polyQ-expanded AR in the adulthood is not sufficient to affect the expression of other ECC genes that instead are dysregulated in the muscle of transgenic mice with constitute expression of polyQ-expanded

AR and in knock-in mice as well as SBMA patients. We added a sentence to the Discussion section relative to this important point.

8. The immunohistochemistry in Fig.4a (for the 8 weeks old WT mice) is not convincing and it appears problematic for the RYR1 immunoreactive signal. The fluorescence signal appears to be diffuse and very weak within the myofibers and nearly equal to that of the background signal which is expected to be nearly black. The authors are encouraged to present better images for this important control. In addition, it appears that all fibers are positive for MyHC type IIb.

To address these concerns, we have repeated the immunofluorescence analysis in transversal sections of quadriceps muscle of WT and SBMA mice both at 4 and 8 weeks of age. We acquired new high-quality pictures from previous experiments and performed new experiments in 3 additional mice for each age and genotype, and we acquired new images. RYR1 staining was homogenous in all the fibers of 4- and 8-week-old WT mice as well as 4-week-old SBMA mice. RYR1 staining was dramatically reduced in 8-week-old mice only in the type II fibers. To be sure that the fiber area lacking RYR1 staining was intact, we stained for the type IIb MyHC. New representative images are shown in Figure 4A.

9. Regarding the experiments on the oxygen consumption rate (OCR), the OCR appears to be lower in the SBMA muscles both in 4 and 8 week old muscles (as shown in both graphs prior to the application by oligomycin). Is this difference statistically significant? The authors did not report any statistical comparison for the Seahorse analysis. Irrespective of the statistical difference, could the authors provide any explanation of the difference in OCR between WT and SBMA muscles?

We apologize for missing this information in the original manuscript. We performed a two-way ANOVA followed by Tukey post-hoc test for all the OCR experiments shown in the revised manuscript. We show that both basal and maximal OCR are reduced in SBMA myofibers compared to WT myofibers. Concerning as to why myofiber respiration is altered in SBMA mice, we added a paragraph (the one before the last paragraph) to the Discussion. Myofiber respiration is altered at presymptomatic stage, likely as the results of multiple factors. We did not detect any defects in circulating lactate, which is increased in conditions when oxidative phosphorylation is impaired, or in the levels of mitochondrial pyruvate carrier. However, we cannot exclude defects in glycolysis or tricarboxylic acid cycle. Concerning the respiratory chain complex activity, we did not detect defects, indicating that myofiber respiration defects do not originate from altered activity of these supercomplexes.

10. The authors state (lines 199-200) “Interestingly, the transcript levels of Nrf1 were downregulated, whereas those of Nrf2, Sdha, and Tfam were upregulated in the skeletal muscle of 8-week-old AR100Q mice compared to control mice (Fig. 4d).” Could the authors expand on the significance of these observations? I fail to understand the importance of the apparent differential regulation of Nrf1 (downregulation) and Nrf2 (upregulation).

We performed analysis of the transcript levels of these genes in the patient-derived specimens (as suggested by reviewer #3), but we could not confirm the data obtained in mice (while the ECC gene expression changes were all

confirmed in patients vs mice). Thus, we removed these data from the manuscript. This did not change the conclusions and data interpretations, rather the revised manuscript is more focused on the analysis of early pathological processes occurring in SBMA muscle.

11. Based on the results shown in Fig.3a regarding Ryr1 mRNA levels, in which AR100Q muscles reveal significant lower levels compared to WT muscles, is not surprising that the authors did not detect any lower immunoreactive signals of Ryr1 (in conjunction with the immunoreactivity against translocase of outer mitochondrial membrane 20 (TOM20), as shown in Fig.6b. Wouldn't the authors expect a significant lower signal of Ryr1 in AR100Q muscles compared to WT muscles? **In the original manuscript, we chose an image that clearly shows RYR signal, but it may seem that RYR1 levels are not decreased, while they are. We performed novel analysis on additional mice and we replaced this image with one that is more representative of the fiber at 12 weeks of age. We thank the reviewer for this comment as this improves the clarity of the manuscript.**

12. The authors should perform statistical testing for the results shown in Fig.7a and reports the results in the graph.

We again thank this reviewer for pointing this out. We performed statistical analysis, as described at point response "9". We found that the OCR of sham-operated SBMA mice was different from that of WT mice and surgically castrated SBMA mice. We apologize for omitting this information in the original manuscript.

Minor concerns

1. In Fig.1d, the authors state that "n=4-6mice/genotype", however in the TETANUS graph, there are only 3 data points in the WT bar. The authors should correct the "n" description.

We apologize for this error. In this experiment, n = 3-6 mice/genotype. Text was corrected accordingly.

2. Details of anesthesia are missing for the CMAP experiments.

We added this information in the Methods section. Mice were anesthetized with Xylazine and Zoletil.

3. It will be very helpful to illustrate with traces the force from a twitch and tetanus in the EDL experiments shown in Fig.1c.

We added this information in Supplementary Figure 1c.

4. The authors should use a different loading control (other than calnexin) in the Westerns presented in Fig.1d, since it is clear that calnexin is not uniform in neither mice or human samples, raising some concerns about the interpretation of the results.

We believe that here the reviewer is referring to Figure 3d. We added all the Western blotting panels showing that although there is some variability among samples, all the comparisons were performed by running the samples in the same SDS-PAGE gels. We cannot use GAPDH as loading control, based on our previous findings⁹. For the sake of clarity, we added all the replicates to Fig.

3d.

Reviewer #3 (Remarks to the Author):

Spinal and bulbar muscular atrophy (SBMA) is a progressive late-onset motor neuron disease caused by abnormal CAG repeat expansions in the androgen receptor (AR) gene. Recent evidences suggest that not only motor neuron but also skeletal muscle are primary contributors to disease pathogenesis in SBMA. Pennuto and her colleague here demonstrated early and late events in SBMA muscles. The main findings of this study are dysregulation of the excitation-contraction coupling (ECC) machinery and mitochondrial dysfunction in the skeletal muscle of SBMA mouse models at an early stage. This finding is important to elucidate the muscle pathophysiology in SBMA. The paper is well written and compelling. The reviewer's concerns are listed below.

Major points:

1. Early defects in mitochondrial respiration in SBMA muscle (Fig. 4), which are reversed by castration (Fig. 7a), are interesting findings of this study. Mitochondrial Ca²⁺ accumulation in the skeletal muscle of 8-week-old AR100Q mice (Fig. 5), which might lead to mitochondrial dysfunction and mitophagy, is also interesting. Confirmation of altered expression of oxidative stress response genes and mitochondrial genes (Figs. 4d and 4e) in patient biopsy samples should strengthen the evidence of these important findings.

We analyzed the transcript levels of these genes in SBMA muscle biopsy specimens and healthy control subjects. However, due to the variability intrinsic to human samples and the relatively low fold change in the transcript levels of these genes, we could not confirm their deregulation in the muscle of SBMA patients. We have previously provided evidence that mitochondria are defective and disposed via mitophagy in the muscle of SBMA patients¹¹. Thus, we decided to remove these RT-PCR data from the manuscript. We are currently collecting more muscle biopsies, and we hope to be able to confirm these data in patients. We believe that this modification does not change the data interpretation, and that rather it increases the focus of the revised manuscript on the cascade of events starting with mitochondrial respiration, contraction dynamics, fatigue and ECC gene deregulation. If the reviewer does feel that we can leave the data in the manuscript, then we are happy to put them back.

2. Muscle contraction/relaxation and force, expression of key excitation-contraction coupling (ECC) genes, and myofiber respiration were significantly affected as early as 4 weeks of age in AR100Q transgenic mice (Figs. 1e, 1f, 3a, 4b), and notably, these defects were consistently observed between 4-week and 8-week-old muscles. Please provide insights into this potential developmental and androgen-independent pathogenesis.

We thank this reviewer for raising this important and intriguing point. Mice reach sexual maturity in between 4 and 6 weeks of age, and testosterone levels rise by 4 weeks of age. To determine whether the ECC deregulation occurs before 4 weeks of age, or in other words, before sexual maturity, we performed RT-PCR analysis in AR100Q mice at 20 days after birth. We did not detect any change in the expression of ECC genes, indicating that their deregulation is tightly linked with androgen-dependent polyQ-expanded AR toxicity. These data are shown in Fig 3a of the revised manuscript. Moreover, we added a

paragraph to the Discussion (last paragraph) section to emphasize that here we identify early pathological processes, namely dysregulation of the ECC process, dynamics of muscle contraction, fatigue and myofiber respiration, that precede motor dysfunction and myofiber disorganization. Nonetheless, they are reverted by androgen deprivation, implying that they occur before onset of motor dysfunction but can be reverted by a therapy that can be started after the onset of these symptoms.

3. The authors demonstrated altered gene expression in sarcomere organization and muscle contraction in pre-symptomatic AR113Q mice (Fig. 2). The authors may want to comment on whether these alterations related to deregulation of ECC machinery or mitochondrial dysfunction.

This is an important point. We added a paragraph to the Discussion section to mention that we cannot establish at this point which one of these two pathological processes is responsible for the subsequent events.

Nonetheless, these processes are tightly linked to each other by Ca²⁺, which acts as a second messenger and influences gene transcription, as well as ATP, which is required by several enzymes involved in metabolism and to ensure muscle contraction/relaxation cycles.

4. Furthermore, some of ECC genes are expressed in a fiber-type specific manner. For instance, Atp2a1 and Casq1 are dominantly expressed in fast-twitch muscles, and Atp2a2 and Casq2 in slow-twitch muscles. Ref 24 showed fast- to slow-twitch muscle fiber-type switching was observed as early as 40-day-old in AR113Q mice. These alterations of ECC genes in the skeletal muscle of SBMA mouse models and patients might be the result of muscle fiber-type switching rather than mutant AR-induced transcriptional change. Indeed, induction of mutant AR by Dox had relatively little effects on expression of Atp2a1 and Casq1 genes (Fig. 3b). This needs some discussion.

We agree with this reviewer that the ECC gene expression changes that we detect early in SBMA muscles might be due at least in part to fiber-type changes. We added this comment to the Discussion section (second paragraph).

Minor points:

1. GO analysis 'cell component' (P5, line 149), 'Biological processes' (line 153), and 'Molecular functions' (line 155) should be 'Cellular component' and 'Biological process', and 'Molecular function', respectively.

We modified the text according to the reviewer indications.

2. On P7, line 223, correct "whichl".

We modified the text according to the reviewer indications. We apologize for this typo.

3. In the upper part of Fig. 8, schematics of SR and mitochondria in physiological condition and SBMA are described, but explanation is lacking. It may be preferable that the authors describe explanation of the schematics in Discussion or the figure legend.

We modified the text (Figure legend) according to the reviewer indications.

Reviewer #4 (Remarks to the Author):

This manuscript is an attempt to describe the time course of events leading to the development of spinobulbar muscular atrophy. To that aim, they used a knock-in mice model showing that altered androgen receptor (AR) alters the generation of muscle force prior to denervation. The pattern of gene expression in the mice model mimics that of tissue from patients with the disease and it is interesting that several genes of the excitation-contraction process are downregulated. These findings correlate with altered morphology of the muscle triad. Another finding is the early alteration of mitochondrial respiration and mitochondria calcium content. The early pathological process can be prevented by AR silencing or surgical castration. The ensemble of results suggests that altered androgen receptors in skeletal muscle participate in a series of transcriptional events that will lead to the development of this disease of which many things remain unknown. This work has an original approach that may facilitate future studies in this interesting area.

We thank this reviewer for these comments.

References

- ¹ Yu, Z. *et al.* Androgen-dependent pathology demonstrates myopathic contribution to the Kennedy disease phenotype in a mouse knock-in model. *J Clin Invest* **116**, 2663-2672, doi:10.1172/JCI28773 (2006).
- ² Lieberman, A. P. *et al.* Peripheral androgen receptor gene suppression rescues disease in mouse models of spinal and bulbar muscular atrophy. *Cell Rep* **7**, 774-784, doi:S2211-1247(14)00087-4 [pii] 10.1016/j.celrep.2014.02.008 (2014).
- ³ Rusmini, P. *et al.* Aberrant Autophagic Response in The Muscle of A Knock-in Mouse Model of Spinal and Bulbar Muscular Atrophy. *Sci Rep* **5**, 15174 (2015).
- ⁴ Polanco, M. J. *et al.* Adenylyl cyclase activating polypeptide reduces phosphorylation and toxicity of the polyglutamine-expanded androgen receptor in spinobulbar muscular atrophy. *Sci Transl Med* **8**, 370ra181 (2016).
- ⁵ Gegenhuber, B. & Tollkuhn, J. Signatures of sex: Sex differences in gene expression in the vertebrate brain. *Wiley Interdiscip Rev Dev Biol* **9**, e348, doi:10.1002/wdev.348 (2020).
- ⁶ Katsuno, M. *et al.* Testosterone reduction prevents phenotypic expression in a transgenic mouse model of spinal and bulbar muscular atrophy. *Neuron* **35**, 843-854, doi:S0896627302008346 [pii] (2002).
- ⁷ Chivet, M. *et al.* Polyglutamine-Expanded Androgen Receptor Alteration of Skeletal Muscle Homeostasis and Myonuclear Aggregation Are Affected by Sex, Age and Muscle Metabolism. *Cells* **9**, doi:cells9020325 [pii] 10.3390/cells9020325 (2020).
- ⁸ Wieters, F., Gruhn, M., Buschges, A., Fink, G. R. & Aswendt, M. Terminal H-reflex Measurements in Mice. *J Vis Exp*, doi:10.3791/63304 (2022).
- ⁹ Rocchi, A. *et al.* Glycolytic-to-oxidative fiber-type switch and mTOR signaling activation are early-onset features of SBMA muscle modified by high-fat diet. *Acta Neuropathol* **132**, 127-144 (2016).
- ¹⁰ Giorgetti, E. *et al.* Rescue of Metabolic Alterations in AR113Q Skeletal Muscle by Peripheral Androgen Receptor Gene Silencing. *Cell Rep* **17**, 125-136 (2016).
- ¹¹ Borgia, D. *et al.* Increased mitophagy in the skeletal muscle of spinal and bulbar muscular atrophy patients. *Hum Mol Genet* **26**, 1087-1103 (2017).

Reviewers' comments:

Reviewer #1 (Remarks to the Author):

The manuscript is improved and addresses my main concerns, though it still does not achieve a substantive mechanistic understanding of the disease process being studied.

In terms of a specific issue that needs to be addressed, there is a problem with the statistical analysis of the data presented in Figure 5. It is incorrect to treat the 64-67 data points in panels a-d as independent measurements, because these data points actually only come from $n = 3$ mice, so it is necessary to average the results per mouse and then perform the t-test analysis. Measurements from the same mouse should never be treated as "independent" for purposes of statistical analysis.

Reviewer #2 (Remarks to the Author):

I have no additional concerns. The authors have covered previous concerns with new data or appropriate analysis and amended textual changes.

Reviewer #3 (Remarks to the Author):

The authors adequately responded to all the issues I raised. I have no further concerns.

Response to reviewer 1 comment 1

In terms of a specific issue that needs to be addressed, there is a problem with the statistical analysis of the data presented in Figure 5. It is incorrect to treat the 64-67 data points in panels a-d as independent measurements, because these data points actually only come from $n = 3$ mice, so it is necessary to average the results per mouse and then perform the t-test analysis. Measurements from the same mouse should never be treated as “independent” for purposes of statistical analysis.

Based on this comment, we performed new statistical analysis of these panels. In detail, to compare Ca^{2+} release from FDB fibers using electrical stimulation between groups, we used linear mixed-effect models (LMM) with the fiber size as the dependent variable, mouse as a random effect, measurement and genotype as fixed-effects, and included a random intercept and random slopes for each measurement for each mouse using the afex package in R (Henrik Singmann, Ben Bolker, Jake Westfall and Frederik Aust 2016). For all tests, the significance threshold was set at $p < 0.05$. We confirmed the conclusions described in the original manuscript. We thank this reviewer for this suggestion, which strengthen the methodology and conclusions of this study.